# De novo activating mutations drive clonal evolution and enhance clonal fitness in *KMT2A*-rearranged leukemia

Axel Hyrenius-Wittsten [1], Mattias Pilheden[1], Helena Sturesson[1], Jenny Hansson [2], Michael P. Walsh[3], Guangchun Song[3], Julhash U. Kazi [4], Jian Liu[1], Ramprasad Ramakrishan[1], Cristian Garcia-Ruiz[1], Stephanie Nance[5], Pankaj Gupta[6], Jinghui Zhang[6], Lars Rönnstrand [4,7,8], Anne Hultquist[9], James R. Downing[3], Karin Lindkvist-Petersson [10], Kajsa Paulsson [1], Marcus Järås[1], Tanja A. Gruber[3,5], Jing Ma[3] & Anna K. Hagström-Andersson [1]

Activating signaling mutations are common in acute leukemia with *KMT2A* (previously *MLL*) rearrangements (*KMT2A*-R). These mutations are often subclonal and their biological impact remains unclear. Using a retroviral acute myeloid mouse leukemia model, we demonstrate that $FLT3^{ITD}$, $FLT3^{N676K}$, and $NRAS^{G12D}$ accelerate *KMT2A-MLLT3* leukemia onset. Further, also subclonal $FLT3^{N676K}$ mutations accelerate disease, possibly by providing stimulatory factors. Herein, we show that one such factor, MIF, promotes survival of mouse *KMT2A-MLLT3* leukemia initiating cells. We identify acquired de novo mutations in *Braf*, *Cbl*, *Kras*, and *Ptpn11* in *KMT2A-MLLT3* leukemia cells that favored clonal expansion. During clonal evolution, we observe serial genetic changes at the $Kras^{G12D}$ locus, consistent with a strong selective advantage of additional $Kras^{G12D}$. *KMT2A-MLLT3* leukemias with signaling mutations enforce *Myc* and *Myb* transcriptional modules. Our results provide new insight into the biology of *KMT2A*-R leukemia with subclonal signaling mutations and highlight the importance of activated signaling as a contributing driver.

[1] Division of Clinical Genetics, Department of Laboratory Medicine, Lund University, 221 84 Lund, Sweden. [2] Division of Molecular Hematology, Department of Laboratory Medicine, Lund University, 221 84 Lund, Sweden. [3] Department of Pathology, St. Jude Children´s Research Hospital, Memphis, TN 38105, USA. [4] Division of Translational Cancer Research, Department of Laboratory Medicine, Lund University, 223 63 Lund, Sweden. [5] Department of Oncology, St. Jude Children´s Research Hospital, Memphis, TN 38105, USA. [6] Department of Computational Biology, St. Jude Children´s Research Hospital, Memphis, TN 38105, USA. [7] Lund Stem Cell Center, Department of Laboratory Medicine, Lund University, 221 84 Lund, Sweden. [8] Division of Oncology, Skane University Hospital, Lund University, 221 85 Lund, Sweden. [9] Department of Pathology, Skane University Hospital, Lund University, 221 85 Lund, Sweden. [10] Medical Structural Biology, Department of Experimental Medical Science, 221 84 Lund University, Lund, Sweden. Correspondence and requests for materials should be addressed to A.K.Höm-A. (email: Anna.Hagstrom@med.lu.se)

Genetic rearrangements of the *Histone-lysine N-methyl-transferase 2A* gene (*KMT2A*, previously *MLL*) is seen in approximately 10% of human acute leukemia[1]. However, in acute lymphoblastic leukemia (ALL) occurring in infants less than 1 year of age, a leukemia subtype with a particularly poor prognosis, close to 80% have *KMT2A*-rearrangements (*KMT2A-R*)[1–3]. We have previously shown that infant *KMT2A*-R ALL has a strikingly low number of mutations with an average of 2.2 non-silent single-nucleotide variants per case, and 1.3 in the dominant leukemia clone[4]. About half of these mutations occurred in kinase/phosphoinositide 3-kinase (PI3K)/RAS signaling pathways, suggesting that they are important cooperating events[4,5]. Kinase/PI3K/RAS pathway mutations are also common in *KMT2A*-R ALL and acute myeloid leukemia (AML) arising in older children and adults[2–4,6,7].

In infant *KMT2A*-R ALL, most of the mutations that deregulate protein components of signal transduction networks are subclonal, as indicated by their mutant allele frequencies (MAF) <0.30. In addition, some cases harbor multiple activating mutations at varying MAFs, suggesting the presence of multiple leukemic clones[4]. Further, analyses of paired diagnostic-relapse samples have shown that signaling mutations may be lost, maintained, or gained at relapse[4,8,9]. Thus, there is a delicate selection process that shapes clonal evolution during leukemia progression, treatment, and relapse. Mouse models have shown a potent genetic cooperativity between signaling mutations and *KMT2A*-Rs, with the most prominent pathophysiological combinatorial gain being acceleration of disease onset[10–17]. However, none of these models have recapitulated the biology of subclonal activating mutations.

The mechanisms by which signaling mutations affect leukemogenesis, whether present at high or low MAF, and the biological processes controlling and shaping the intricate selection of evolving leukemic clones, remain poorly understood. To gain a better understanding of these processes, we studied the functional importance of constitutively activated FLT3- and RAS-signaling in *KMT2A*-mediated leukemogenesis in mouse. We focused our efforts on $FLT3^{ITD}$, $FLT3^{N676K}$, and $NRAS^{G12D}$, as *FLT3* and *NRAS* are the most common targets of mutations that deregulate signal transduction in AML, and studied clonal evolution of leukemia cells carrying subclonal activating mutations over time.

Herein, we show that also subclonal activating mutations, as demonstrated by $FLT3^{N676K}$, accelerate *KMT2A*-*MLLT3* leukemia onset. Moreover, we identify acquired de novo activating mutations in known cancer-associated genes in *KMT2A*-*MLLT3* leukemia cells that favored clonal expansion, indicating a strong

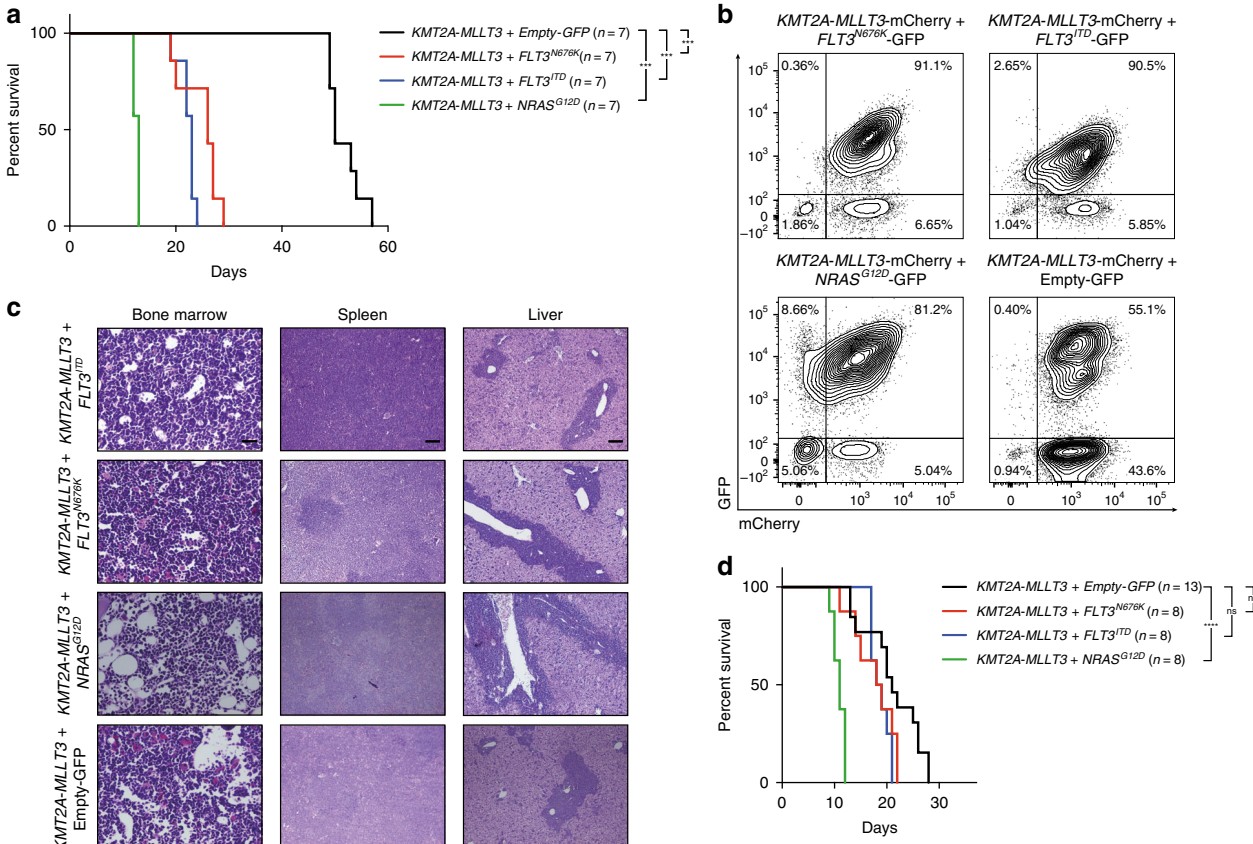

**Fig. 1** FLT3- and RAS-signaling mutations accelerate AML onset. **a** Kaplan–Meier survival curves of mice transplanted with bone marrow (BM) cells co-transduced with *KMT2A*-*MLLT3* and either $FLT3^{ITD}$, $FLT3^{N676K}$, or $NRAS^{G12D}$ ($n = 7$ for all groups), showing accelerated disease onset for mice receiving *KMT2A*-*MLLT3* and either of the activating mutations ($P = 0.0002$ for *KMT2A*-*MLLT3* + $FLT3^{ITD}$, $P = 0.0002$ for *KMT2A*-*MLLT3* + $FLT3^{N676K}$, and $P = 0.0004$ for *KMT2A*-*MLLT3* + $NRAS^{G12D}$, as compared to *KMT2A*-*MLLT3* + Empty-GFP. Mantel–Cox log-rank test). **b** Flow cytometric analysis of BM from sacrificed mice showed that a majority of cells were GFP⁺mCherry⁺ in mice co-expressing *KMT2A*-*MLLT3* and an activating mutation.
**c** Hematoxylin–eosin stained sections from bone marrow, liver, and spleen (original magnification 200×, scale bar 0.1 mm, for bone marrow and 40×, scale bar 0.5 mm, for spleen and liver). The architecture of the spleen is effaced and the red pulp is expanded mainly due to expansion of immature myeloid cells. In the liver, periportal, perisinusoidal and intrasinusoidal extensive infiltrates of immature hematopoietic cells were noted. **d** Kaplan–Meier curves for secondary recipients transplanted with primary leukemic splenocytes showing that only *KMT2A*-*MLLT3* + $NRAS^{G12D}$ sustained a significant difference in disease latency when compared to *KMT2A*-*MLLT3* + Empty-GFP ($P < 0.0001$. Mantel–Cox log-rank test). ***$P ≤ 0.001$, ****$P ≤ 0.0001$; ns, not significant

selective pressure for activated signaling as a cooperating event in KMT2A-R leukemogenesis.

## Results

**FLT3- and RAS-signaling mutations accelerate AML onset**. We here studied constitutively activated FLT3- and RAS-signaling in KMT2A-R leukemogenesis. First, we focused on the impact of $FLT3^{ITD}$, $FLT3^{N676K}$, or $NRAS^{G12D}$ when present as a dominant leukemia clone at disease manifestation. Mouse hematopoietic stem and progenitor cells (c-Kit$^+$ cells) were co-transduced with retroviral vectors expressing KMT2A-MLLT3-mCherry and $FLT3^{ITD}$-GFP, $FLT3^{N676K}$-GFP, $NRAS^{G12D}$-GFP, or a GFP control vector (Empty-GFP) (Supplementary Fig. 1a, b). Expression of mutant NRAS and FLT3 in the mouse leukemia cell line Ba/F3 resulted in elevated phosphorylated p-AKT and p-ERK1/2, as well as p-STAT5 for $FLT3^{ITD}$ (Supplementary Fig. 1c).

Transduced cells were transplanted unfractionated and at transplantation, only a minor fraction of cells co-expressed KMT2A-MLLT3 and $NRAS^{G12D}$, $FLT3^{ITD}$, or $FLT3^{N676K}$ (Supplementary Fig. 1d). Despite this, all of them accelerated leukemia onset (median latency of 13, 23, 26 days, respectively, versus 50 days for KMT2A-MLLT3 alone; $P = 0.0004$, $P = 0.0002$, and $P = 0.0002$, respectively. Mantel–Cox log-rank test) (Fig. 1a), consistent with a competitive growth advantage for these cells. In agreement with this, a majority of cells in the bone marrow (BM) contained both a signaling mutation and KMT2A-MLLT3 (Fig. 1b, Supplementary Fig. 1e), supporting a prominent genetic

cooperativity between the KMT2A-R and activating FLT3 and RAS mutations, including the recently identified $FLT3^{N676K}$ [4,18]. Mice displayed leukocytosis, splenomegaly, expression of CD11b and Gr-1 in BM cells, and a similar burden of leukemic granulocyte-macrophage-like-progenitors[10,19] (Supplementary Fig. 1f–i and Supplementary Fig. 2a–c). The BM, spleen, and liver showed infiltration of blasts and cytopenia, and peripheral blood (PB) an increase of blasts and progenitors (>20%) (Fig. 1c). Thus, mice succumbed to AML[20]. Leukemias with an activating mutation showed presence of p-ERK1/2, and also p-STAT5 for those with $FLT3^{ITD}$. All leukemias displayed similar phosphorylation levels of P38 (Supplementary Fig. 3).

Leukemic cells isolated from the spleen of moribund mice gave rise to secondary malignancies identical to the primary disease in sublethally irradiated recipients, and with significantly reduced disease latency (Fig. 1d and Supplementary Fig. 4a–d). A continued enrichment for KMT2A-R leukemic cells carrying $FLT3^{ITD}$, $FLT3^{N676K}$, or $NRAS^{G12D}$, was seen, consistent with a persistent competitive advantage for these cells (Supplementary Fig. 4e). Disease latency of secondary KMT2A-MLLT3 recipients was similar to those with an activating mutation, suggesting that they had adapted a comparable leukemic phenotype to those with activating mutations.

**Subclonal $FLT3^{N676K}$ accelerates AML onset**. In infant KMT2A-R ALL, a majority of the kinase/PI3K/RAS-pathway mutations are present in subclones[4,5], raising the possibility that cells with

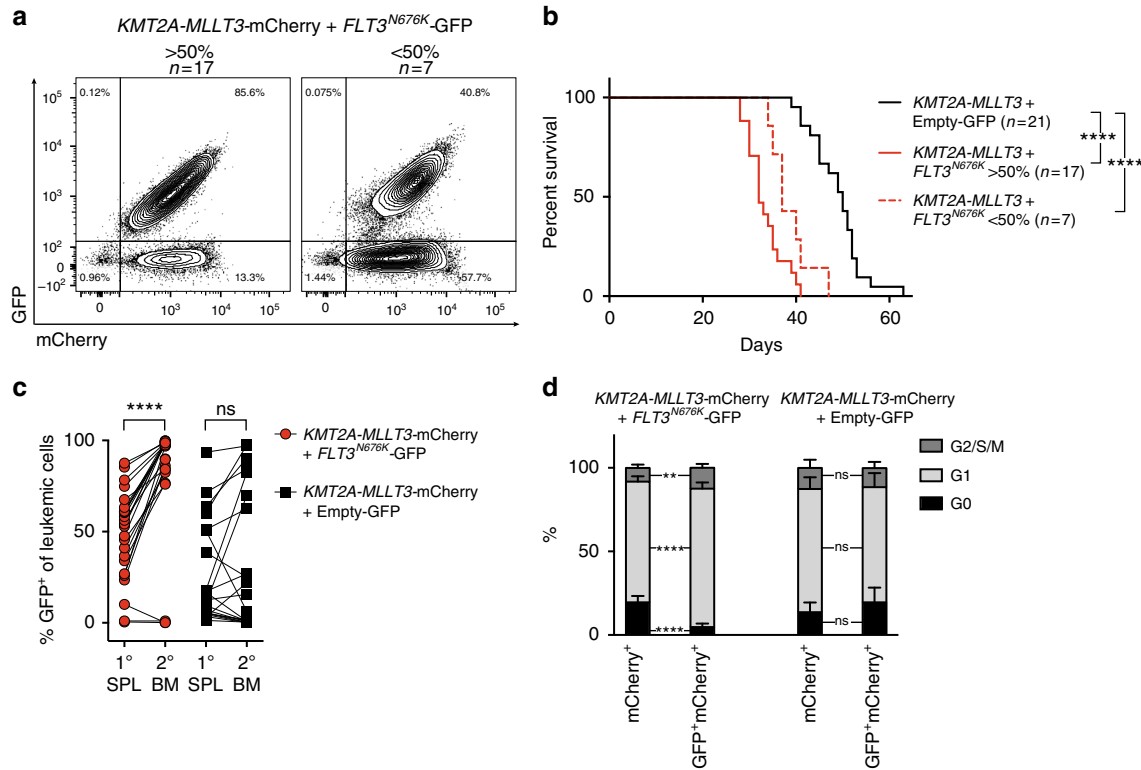

**Fig. 2** Subclonal $FLT3^{N676K}$ accelerates AML onset. **a** Flow cytometric analysis on BM cells from primary KMT2A-MLLT3 + $FLT3^{N676K}$ recipients revealed the presence of either a dominant clone (>50% GFP$^+$mCherry$^+$, $n = 17$), or a subclone (<50% GFP$^+$mCherry$^+$, $n = 7$) of KMT2A-MLLT3-mCherry + $FLT3^{N676K}$-GFP expressing cells. **b** Kaplan–Meier curves of mice transplanted with KMT2A-MLLT3 and $FLT3^{N676K}$ ($n = 24$) or with Empty-GFP vector control ($n = 21$) showing accelerated disease onset for both dominant and subclonal KMT2A-MLLT3 + $FLT3^{N676K}$ leukemias as compared to KMT2A-MLLT3 + Empty-GFP (both $P < 0.0001$. Mantel–Cox log-rank test). **c** Evolution of $FLT3^{N676K}$-GFP$^+$ cells within the mCherry$^+$ leukemic population between primary (1°) spleen (SPL) and secondary (2°) BM showed a significant expansion of KMT2A-MLLT3-mCherry + $FLT3^{N676K}$-GFP cells ($n = 24$, $P < 0.0001$. Paired t-test) but not of KMT2A-MLLT3-mCherry + Empty-GFP cells ($n = 21$, $P = 0.1468$. Paired t-test). **d** Cell cycle analysis of primary leukemia cells showed a higher cell cycle rate for KMT2A-MLLT3-mCherry cells harboring $FLT3^{N676K}$-GFP ($n = 7$) as compared to KMT2A-MLLT3 + Empty-GFP ($n = 7$). Error bars are s.d. **P ≤ 0.01, ****P ≤ 0.0001; ns, not significant

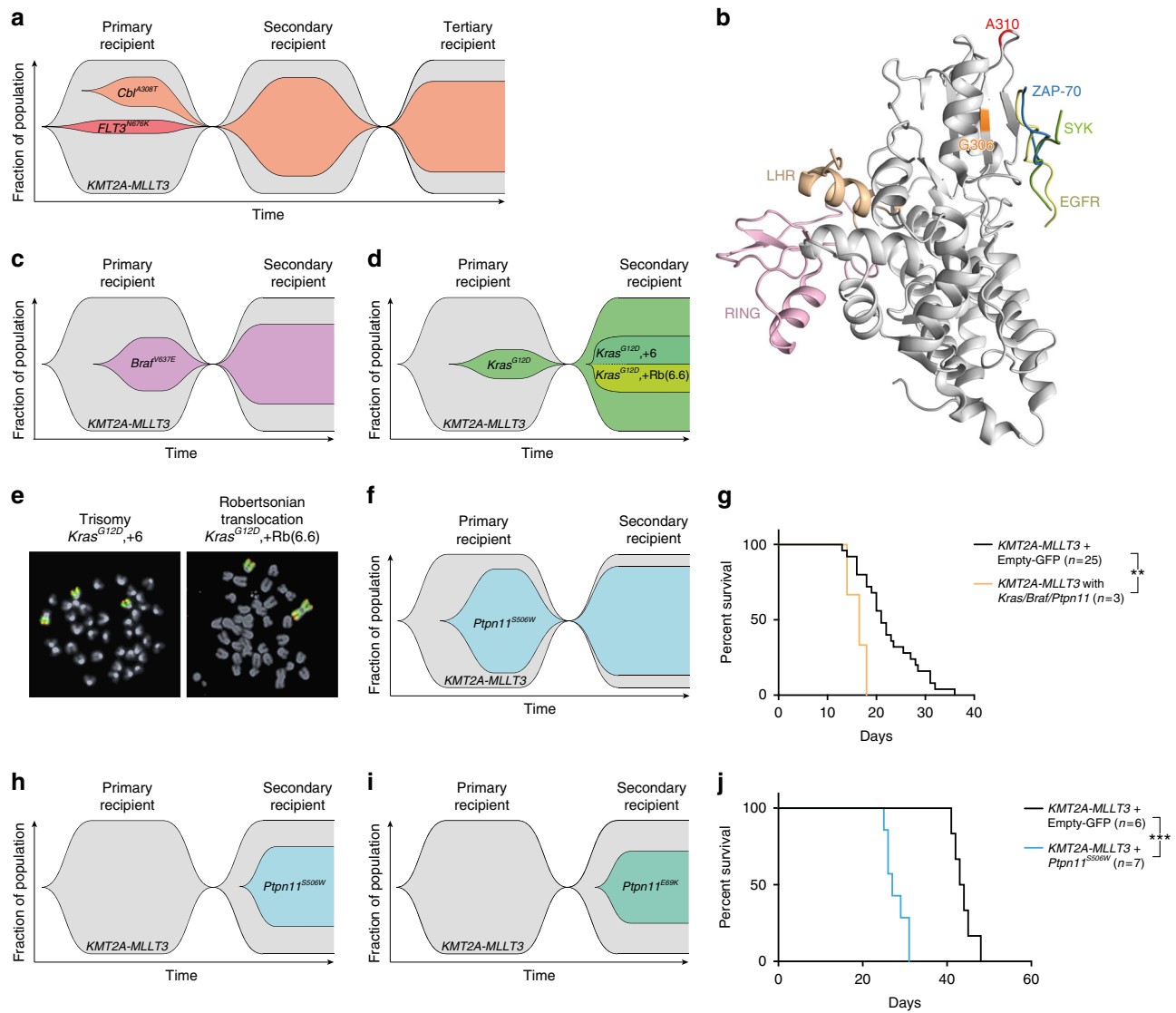

**Fig. 3** *KMT2A-MLLT3* cells acquire de novo signaling mutations. **a** Fish plot showing progression of one leukemia with a subclonal *KMT2A-MLLT3* + *FLT3^N676K* (10.1%) that gained a *Cbl^A308T* de novo mutation in the primary (SJ016337) *KMT2A-MLLT3*-only cells which expanded and gained clonal dominance a secondary (SJ046291), and tertiary recipient; MAF 0.11->0.37->0.34. **b** Ribbon representation of the human N-terminal SH2-containing tyrosine kinase-binding (TKB) domain of CBL (gray), overlayed with ZAP-70 (blue), SYK (green) and EGFR (yellow) peptides, and including the RING domain (pink), LHR (beige), and with residue G306 (orange) and A310 (red) highlighted. **c**, **d** Fish plots of primary *KMT2A-MLLT3* + Empty-GFP recipients that gained de novo mutations and their subsequent progression in secondary recipients showing **c** a primary subclonal *Braf^V637E* that increased in size; MAF 0.20 (SJ018146) ->0.30 (SJ046293), **d** a subclonal *Kras^G12D* that progressed to clonal dominance; MAF 0.11 (SJ016338) ->0.59 (SJ046295). The MAF of 0.59 indicated allelic imbalance and was caused by a gain of chromosome 6 (+6) and a Robertsonian translocation (+Rb(6.6)), each present in 20% of cells, as shown by fluorescence in situ hybridization (FISH). **e** FISH results of the *Kras* locus (red) in **d**, showing gain of chromosome 6 (green) by trisomy in 8/38 (21%) metaphases and by a Robertsonian translocation involving two copies of chromosome 6 (Rb(6.6)) in addition to a normal chromosome 6 in 8/38 (21%) metaphases. Together with the observed MAF this suggests that all disomic cells were heterozygous for *Kras^G12D* mutation and all cells with three copies of chromosomes 6 had duplicated the mutated allele. **f** Fish plot showing a dominant *Ptpn11^S506W* which was preserved in size; MAF 0.39 (SJ016332) ->0.41 (SJ046294). **g** Kaplan–Meier survival curve for secondary *KMT2A-MLLT3* recipients, showing accelerated disease for mice that harbored *Braf^V637E*, *Kras^G12D*, and *Ptpn11^S506W* (n = 3) as compared to those lacking an identified de novo mutation (n = 25; P = 0.0083. Mantel–Cox log-rank test). **h**, **i** Fish plots of secondary *KMT2A-MLLT3* + Empty-GFP recipients that gained de novo mutations in **h** *Ptpn11^S506W* and **i** *Ptpn11^E69K* (MAF 0.31 and 0.29, respectively). **j** Kaplan–Meier curves of mice transplanted with BM co-transduced with either *KMT2A-MLLT3* + *Ptpn11^S506W* (n = 7) or *KMT2A-MLLT3* + Empty-GFP (n = 6), showing accelerated disease on onset for *KMT2A-MLLT3* + *Ptpn11^S506W* recipients (P = 0.0005. Mantel–Cox log-rank test). **P ≤ 0.01, ***P ≤ 0.001

activating mutations could affect the growth or survival of other leukemia cells. We hypothesized that a low initial number of cells co-expressing an activating mutation and *KMT2A-MLLT3* together with a several fold higher number of cells expressing only *KMT2A-MLLT3* would allow for the formation of distinct subclones and performed three additional experiments accordingly

(1:28, 1:41, and 1:156 for *KMT2A-MLLT3* + *FLT3^N676K*:*KMT2A-MLLT3* alone, respectively) (Supplementary Fig. 5a). We focused our analyses on *FLT3^N676K*, the most common *FLT3* mutation in our series of infant and childhood *KMT2A*-R leukemia[4].

Combined analysis of the transplantations showed that despite the low number of cells that co-expressed *KMT2A-MLLT3* and

$FLT3^{N676K}$ at transplantation, these mice displayed accelerated AML onset (median latency 34 versus 50 days for $KMT2A$-$MLLT3$ alone, $P < 0.0001$. Mantel–Cox log-rank test) (Supplementary Fig. 5b-f). Interestingly, the BM of $KMT2A$-$MLLT3$ + $FLT3^{N676K}$ recipients were comprised of either (1) a dominant clone, >50% (17/24 mice), or (2) a subclone, <50% (7/24 mice) of $KMT2A$-$MLLT3$ + $FLT3^{N676K}$ expressing cells, hereafter referred to as dominant clone and subclone, respectively (Fig. 2a). Importantly, when recipients were divided based on the fraction of $FLT3^{N676K}$ expressing leukemic cells, mice with subclonal $FLT3^{N676K}$ still succumbed to disease at an earlier onset (dominant clone, 32 days, $P < 0.0001$; subclone, 37 days $P < 0.0001$. Mantel–Cox log-rank test) (Fig. 2b). There was a weak but significant correlation between clone size and disease latency ($r_s$ = −0.586, $P = 0.0026$. Spearman´s rank correlation coefficient) (Supplementary Fig. 5g). Our data therefore suggest that subclonal activating mutations accelerate $KMT2A$-R leukemogenesis.

**Clonal evolution of $FLT3^{N676K}$ containing leukemic cells.** We next investigated the evolution of cells co-expressing $FLT3^{N676K}$ and $KMT2A$-$MLLT3$ in secondary sublethally irradiated recipients. Primary splenocytes were used for these experiments and the fraction of cells co-expressing $FLT3^{N676K}$ and $KMT2A$-$MLLT3$ was significantly smaller in spleen as compared to BM with 13/24 mice having a dominant clone and 11/24 a subclone (Supplementary Fig. 5h). Disease latency in secondary recipients was reduced and a small difference in survival was observed; median 16 days for $KMT2A$-$MLLT3$ + $FLT3^{N676K}$ and 21 days for $KMT2A$-$MLLT3$ alone, $P = 0.0293$ (Mantel–Cox log-rank test) (Supplementary Fig. 5i). Secondary recipients were then divided based on the fraction of $FLT3^{N676K}$ containing cells in the spleens of primary recipients, showing that size had a modest effect on disease latency (Supplementary Fig. 5j, k).

Clonal evolution of leukemia splenocytes from primary recipients in the BM of secondary recipients showed three distinct patterns. Most frequently, the $KMT2A$-$MLLT3$ + $FLT3^{N676K}$ cells increased in size (21/24 mice) (Fig. 2c and Supplementary Fig. 5l). In concordance, $FLT3^{N676K}$ expression induced a higher cell cycle rate (Fig. 2d). However, mice receiving cells from primary recipients with a very small $KMT2A$-$MLLT3$ + $FLT3^{N676K}$ clone (≤10.1%, 3/24 mice) displayed a diverse evolution where one subclone (1.1%) was maintained in size, one (0.5%) disappeared, and one decreased in size from 10.1% to 0.1 % (Fig. 2c and Supplementary Fig. 5m, n). Thus, cells containing an activating mutation typically had a persistent selective advantage also in established disease. However, in a few recipients, these cells failed to outgrow leukemia cells containing $KMT2A$-$MLLT3$ alone (see below).

**$KMT2A$-$MLLT3$ cells acquire de novo signaling mutations.** The failure of $FLT3^{N676K}$ containing cells to expand in a subset of secondary recipients suggested that the $KMT2A$-$MLLT3$ only cells could have acquired de novo mutations. We therefore sequenced 41 selected genes that were either tyrosine kinases, within the PI3K/RAS pathways, epigenetic regulators, or recurrently mutated in $KMT2A$-R leukemia (see Methods and Supplementary Table 1) at a mean coverage of $3231 \pm 1277$ (s.d.) (Supplementary Data 1), in 62 primary (27 dominant and 7 subclonal $KMT2A$-$MLLT3$ and an activating mutation, and 28 $KMT2A$-$MLLT3$ alone) and 29 paired secondary recipients (derived from 15 dominant and 6 subclonal $KMT2A$-$MLLT3$ and an activating mutation, and 8 $KMT2A$-$MLLT3$ alone) (Supplementary Table 2). Notably, our analyses identified acquired de novo mutations in RAS pathway genes in 4/62 primary- and in 6/29 paired

secondary recipients, all of which occurred in genes known to be mutated in human $KMT2A$-R leukemia ($Ptpn11$, $Cbl$, $Braf$, $Kras$) and all but one mutation ($Cbl^{A308T}$) occurred at amino acid positions known to be activating (Supplementary Fig. 6a-h and Supplementary Data 2)[21–24]. In every case, as determined by RNA sequencing, the acquired mutation was expressed (Supplementary Data 2).

Only 1/34 (3%) of the primary $KMT2A$-$MLLT3$ recipients that co-expressed $FLT3^{ITD}$, $FLT3^{N676K}$, or $NRAS^{G12D}$ had acquired a de novo mutation ($Cbl^{A308T}$) (Fig. 3a and Supplementary Data 2). Strikingly, $Cbl^{A308T}$ was detected in the mouse in which the frequency of $KMT2A$-$MLLT3$ + $FLT3^{N676K}$ cells decreased in size in the secondary recipient. The $Cbl^{A308T}$ was subclonal in the primary (MAF 0.11), dominant in the secondary (MAF 0.37) and maintained in size in a tertiary recipient (Fig. 3a and Supplementary Data 2). Thus, in the primary recipient, $KMT2A$-$MLLT3$ cells acquired a $Cbl^{A308T}$ that had a selective advantage over $FLT3^{N676K}$ leukemia cells and gained clonal dominance in the secondary recipient. $Cbl$ encodes a RING ubiquitin ligase that functions as a negative regulator of signaling pathways. The corresponding human residue of A308 (A310) is located in the highly conserved N-terminal SH2 domain within the tyrosine kinase-binding domain (Fig. 3b and Supplementary Fig. 6a, b), which mediates substrate specificity and is critical for substrate ubiquitination. Mutations occur over the whole region of $CBL$, with the RING domain being a hotspot; no A310 mutations have been reported in COSMIC[25] (Supplementary Fig. 6b). The $CBL^{G306E}$ results in a complete loss of its ability to bind SYK and FLT3[26,27]. Given the close proximity of A310 and G306 (Fig. 3b and Supplementary Fig. 6i), $Cbl^{A308T}$ likely also interferes with binding and subsequent ubiquitination of target substrates, such as Syk and Flt3. Mutations in $CBL$ cause factor independent growth of the IL3 dependent cell line Ba/F3 when overexpressed with certain receptor kinases such as $FLT3$[28]. Using the same approach[28–30], we functionally demonstrated that expression of $Cbl^{A308T}$ resulted in a significantly higher cell proliferation as compared to $Cbl^{WT}$, supporting that the A308T mutation interferes with the ability of Cbl to negatively regulate intracellular signaling (Supplementary Fig. 7a). Taken together, when the constitutively active mutation is present as a subclone, there is a selective benefit of acquiring activating mutations in leukemia cells lacking such a mutation.

Acquired mutations were more common in mice lacking a constitutively expressed activating mutation, with 3/28 (11%) primary and 5/8 secondary $KMT2A$-$MLLT3$ recipients containing such mutations. Mutations in $Braf^{V637E}$, $Kras^{G12D}$, and $Ptpn11^{S506W}$ were identified in three primary and in their corresponding secondary recipient (Fig. 3c–f and Supplementary Data 2). In secondary transplantations, the three $KMT2A$-$MLLT3$ leukemias with $Braf^{V637E}$, $Kras^{G12D}$, or $Ptpn11^{S506W}$ were associated with accelerated disease (Fig. 3g). Further, a $Ptpn11^{S506W}$ and a $Ptpn11^{E69K}$ were identified in two additional secondary recipients, with no evidence of the mutant allele in their corresponding primary recipient (Fig. 3h, i and Supplementary Data 2). Using directed amplicon sequencing, additional secondary $KMT2A$-$MLLT3$ leukemias were screened for $Braf^{V637E}$ ($n = 14$), $Cbl^{A308T}$ ($n = 14$), $Kras^{G12D}$ ($n = 15$), $Ptpn11^{E69K}$ ($n = 15$), and $Ptpn11^{S506W}$ ($n = 13$), but no mutations were identified. The $Braf^{V637E}$ and $Kras^{G12D}$ (human orthologues $BRAF^{V600E}$ and $KRAS^{G12D}$) are well-described mutations that cause constitutively active signaling in humans and mice (Supplementary Fig. 6c–f)[21,22]. The only recurrently mutated gene was the proto-oncogene tyrosine phosphatase $Ptpn11$ (encoding Shp-2), acquired in three separate leukemias, $Ptpn11^{E69K}$ ($n = 1$) and $Ptpn11^{S506W}$ ($n = 2$). These mutations are located in the C-terminal SH2 and protein tyrosine

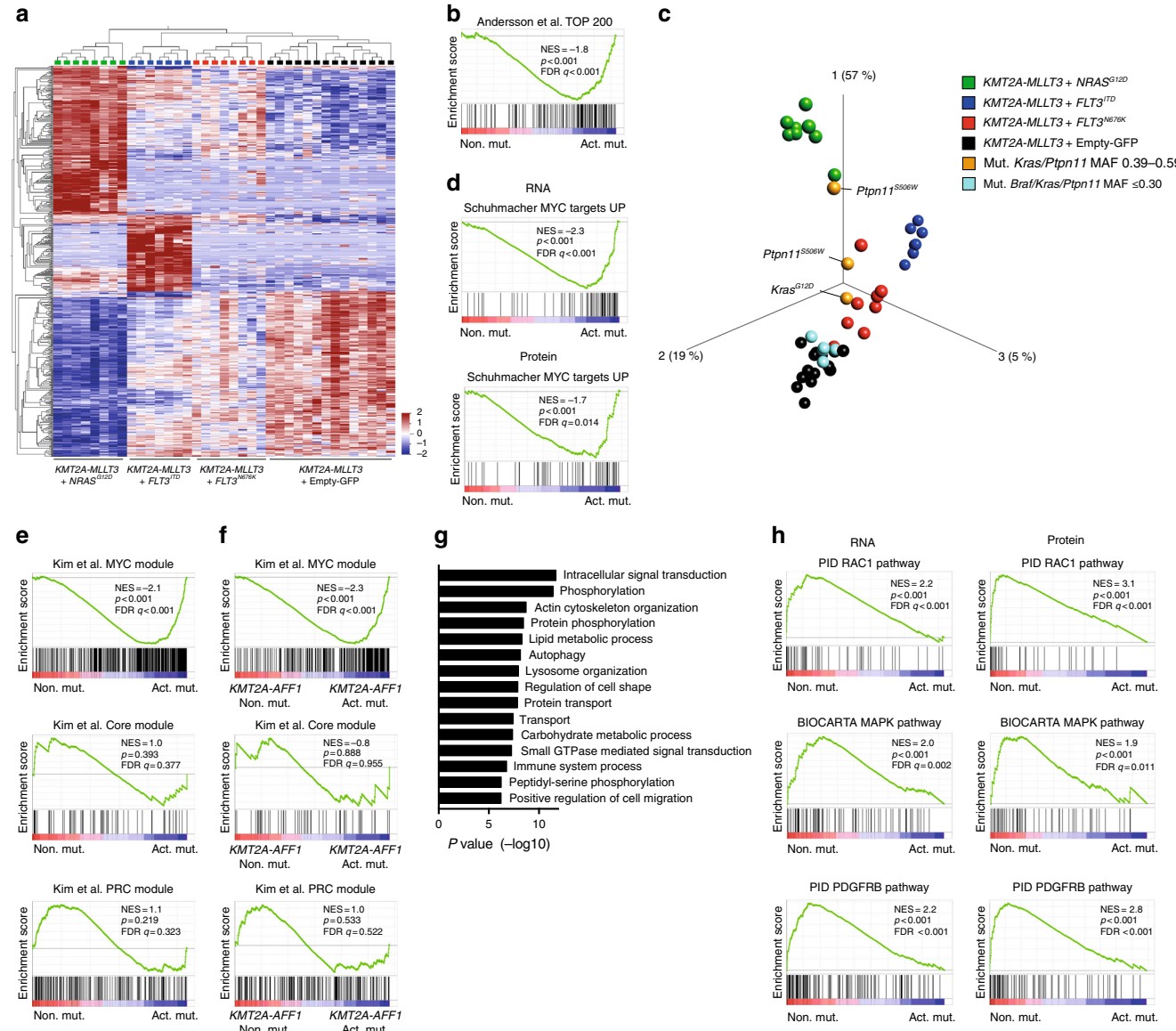

**Fig. 4** Activating mutations enforce *Myc* and *Myb* modules. **a** Hierarchical clustering after multigroup comparison of mouse leukemias using 553 variables ($P = 2.9\mathrm{e}^{-8}$, FDR $= 6.6\mathrm{e}^{-7}$, *F*-test) reveals that each activating mutation causes a distinct gene expression profile. **b** GSEA revealed enrichment of a signature of the top 200 genes that discriminate infant *KMT2A-AFF1* ALL patients harboring activating mutations (dominant or subclonal) to those lacking such mutations[4] for mouse leukemias carrying *KMT2A-MLLT3* with *FLT3*[ITD], *FLT3*[N676K], or *NRAS*[G12D] (Act. mut.). **c** Principal component analysis (553 variables, $P = 2.9\mathrm{e}^{-8}$, FDR $= 6.6\mathrm{e}^{-7}$, *F*-test) of primary mouse leukemias. *KMT2A-MLLT3* leukemias with de novo mutations in *Braf*, *Kras* or *Ptpn11* were inserted into the same PCA (still based solely the leukemias defined above), showing that leukemias with de novo mutations with a MAF of 0.39–0.59 now fall closely to leukemias co-expressing *KMT2A-MLLT3* and *FLT3*[ITD], *FLT3*[N676K], or *NRAS*[G12D]. **d** GSEA revealed enrichment of a MYC signature in the presence of *FLT3*[ITD], *FLT3*[N676K], or *NRAS*[G12D] (Act. mut.) both at the RNA and protein level. **e** GSEA revealed enrichment of the MYC module, but not Core or polycomb (PRC) modules[38], for mouse *KMT2A-MLLT3* leukemias with *FLT3*[ITD], *FLT3*[N676K], or *NRAS*[G12D] (Act. mut.). **f** GSEA revealed enrichment of the MYC module, but not Core or PRC modules[38], in infant *KMT2A-AFF1* ALL patients with activating mutations (dominant or subclonal; Act. mut.) to those lacking such mutations (Non. mut.). **g** Gene ontology enrichment results of genes upregulated in leukemias *KMT2A-MLLT3* + Empty-GFP as compared to *KMT2A-MLLT3* leukemias with either *FLT3*[ITD], *FLT3*[N676K], or *NRAS*[G12D] (FDR ≤0.01). **h** GSEA on transcriptomic- and proteomic data showed enrichment of intracellular signaling pathways, such as RAC1, MAPK, and PDGFRB, in *KMT2A-MLLT3* leukemias lacking an activating mutation

phosphatase catalytic domains, known hotspots in human disease (E69 and S502) (Supplementary Fig. 6g, h). SHP-2 positively regulate RAS signaling[31] and mutations are seen in *KMT2A-R* acute leukemia of all ontogenies[4,7]. Mutations at E69 and S502 are known to increase SHP-2 phosphatase activity[23,24], suggesting that also the mouse mutations would have similar effect. Co-expression of *Ptpn11*[S506W] and *KMT2A-MLLT3* accelerated AML onset compared to control mice (median latency 27 vs. 44 days, $P = 0.0005$. Mantel–Cox log-rank test) with a majority of *KMT2A-*

*MLLT3* cells containing *Ptpn11*[S506W] (Fig. 3j and Supplementary Fig. 7b–f). Thus, *Ptpn11*[S506W] is a cooperating mutation in *KMT2A-R* leukemogenesis.

The availability of a paired secondary leukemia allowed us to study the evolution of the acquired mutations. The *Braf*[V637E] was subclonal at leukemia onset (MAF 0.20) and increased in size in the secondary recipient (MAF 0.30) (Fig. 3c and Supplementary Data 2). Similarly, *Kras*[G12D] was subclonal (MAF 0.11) at leukemia manifestation and gained clonal dominance in the

                                                                 

secondary recipient (MAF 0.59) (Fig. 3d and Supplementary Data 2). The high MAF of 0.59 indicated allelic imbalance at the locus. Fluorescence in situ hybridization (FISH) analysis showed gain of one additional copy of chromosome 6, where *Kras* is located, by two different mechanisms in 40% of the nuclei in the secondary recipient, agreeing well with the increased MAF and suggesting a strong selective advantage of additional *Kras*[G12D] (Fig. 3e and Supplementary Table 3). Finally, *Ptpn11*[S506W] was present in a majority of cells at leukemia onset (MAF 0.39) and was maintained in size in the secondary recipient (MAF 0.41) (Fig. 3f and Supplementary Data 2). Taken together, in this mouse model, there is a strong selective pressure for activated signaling as a cooperating event in *KMT2A*-R leukemogenesis and in the absence of a constitutively active signaling mutation, cells acquire spontaneous de novo mutations in genes that promote active signaling. Typically, these mutations altered clonal fitness and favored clonal outgrowth, consistent with that they confer a competitive advantage.

**Activating mutations enforce *Myc* and *Myb* modules.** The transcriptional landscape of *KMT2A*-R leukemia has been comprehensively studied in mice[19,32–35] however, insight into the effect of an activating mutation is limited[33]. We therefore performed RNA sequencing on 45 mouse leukemias (Supplementary Data 3) and also included data from purified normal immature hematopoietic subpopulations from the mouse (Supplementary Table 4)[36]. In addition, to study effects induced by activating mutations at the protein level, we performed mass spectrometry (MS)-based quantitative proteomic analysis by stable isotope labeling on 15 leukemias (Supplementary Data 3, 4).

Principal component analysis of purified normal hematopoietic subpopulations showed distinct clusters based on maturation stage and that the leukemias mostly resembled normal granulocyte–macrophage progenitors (GMPs) (Supplementary Fig. 8a)[19]. All leukemias had high expression of *Hoxa9*, *Hoxa10*, and *Meis1*, known *KMT2A*-R target genes, as compared to GMP cells (Supplementary Fig. 8b)[19]. Gene set enrichment analysis (GSEA)[37] revealed enrichment of signatures from pediatric *KMT2A*-R AML (Supplementary Fig. 8c and Supplementary Data 5), supporting that mouse leukemias activate similar pathways as human *KMT2A*-R leukemia and is a representative model of this disease.

Despite a shared *KMT2A*-R leukemic signature, mutant *FLT3* and *NRAS* expression also induced distinct gene expression profiles (GEPs) (Fig. 4a, Supplementary Fig. 8d–f and Supplementary Data 6). To link the GEPs induced by activating mutations in our mouse model to that seen in human disease, GSEA was performed using a gene signature associated with infants harboring *KMT2A*-AFF1 and an activating mutation[4], which demonstrated enrichment of this signature in mice with activating mutations (Fig. 4b). This indicates that the transcriptional changes induced by activating mutations in human disease are preserved in the mouse. Notably, identified de novo mutations influenced the GEP for samples with mutations in the dominant leukemia clone, i.e., *Kras* and *Ptpn11* (MAF 0.39–0.59), now clustering closely to those with a constitutively expressed mutation; this was not seen for mutations with MAF ≤ 0.30 (Fig. 4c).

Although *FLT3*[ITD], *FLT3*[N676K], and *NRAS*[G12D] each had distinct GEPs, they induced similar biological processes at the gene and protein level, including a distinct enrichment of MYC signatures (Fig. 4d and Supplementary Data 7–14). The MYC regulatory network enhances embryonic stem cell (ESC) programs, but is separate from the pluripotency network[38]. Indeed, the signature evoked by activating mutations was specific

to the MYC module[38] and not linked to ESC self-renewal, given that no enrichment of the core and polycomb modules[38] was observed (Fig. 4e, Supplementary Fig. 8g, h, and Supplementary Data 15–23). In line with this, ribosomal and translational signatures, MYC influenced processes[39], were enriched at the RNA and protein level (Supplementary Fig. 8i–k and Supplementary Data 24–26). The MYC signature likely reflect a worse prognosis[38,40], in line with the rapid disease progression for these mice[34,38,41] (Supplementary Fig. 9a and Supplementary Data 16–19). *Myc* is a direct target gene of MYB[42], a transcription factor known to be a key regulator of hematopoietic stem and progenitor cells and to be an important mediator in *KMT2A*-R leukemia[34,35]. Further, *Myb* is associated with a leukemia maintenance signature in *KMT2A*-R leukemia models[35] and this signature was enriched in leukemias with an activating mutation at the gene and protein level (Supplementary Fig. 9b, c and Supplementary Data 16–23). However, *Myb* was not part of the leading-edge genes, but was uniformly expressed in all leukemias, including those expressing *KMT2A-MLLT3* alone Supplementary Fig. 9b–d, and Supplementary Data 17–19). The *Myc* and *Myb* signatures mainly associated with committed progenitors as opposed to regenerative hematopoietic populations (Supplementary Fig. 9e, f). Enrichment of the *Myc* and *Myb* signatures was also observed in mouse leukemias with de novo *Kras/Ptpn11* mutations (MAF 0.39–0.59) and in primary infant *KMT2A-AFF1* ALL harboring activating mutations[4] (Fig. 4f, Supplementary Fig. 9g-j, and Supplementary Data 27, 28).

To gain further insight into the transcriptional effect on downstream MEK/ERK signaling in leukemias with activating mutations, we selectively investigated if expression of *FLT3*[ITD], *FLT3*[N676K], and *NRAS*[G12D] caused increased expression of known transcriptional output genes and negative feedback regulators of MEK/ERK signaling[43]. Indeed, enrichment of these genes were particularly strong in *KMT2A-MLLT3* mouse leukemia with *NRAS*[G12D] and in infant *KMT2A-AFF1* ALL patients harboring activating mutations (Supplementary Fig. 9k and Supplementary Data 19, 28)[4]. However, enrichment was also seen for *KMT2A-MLLT3* mouse leukemia with *FLT3*[N676K] or *Kras/Ptpn11* mutations (MAF 0.39–0.59) (Supplementary Fig. 9k and Supplementary Data 18, 27). This implicates an increased transcriptional activity of the MEK/ERK signaling pathway including negative feedback regulators in leukemia with signaling mutations, in particular those with *FLT3*[N676K] and *NRAS*[G12D], and that the mutant cells are insensitive to negative feedback control. The same enrichment was not observed for *FLT3*[ITD] leukemias (Supplementary Fig. 9k and Supplementary Data 17).

*KMT2A-MLLT3* recipients lacking an activating mutation had increased expression of genes involved in signal transduction (Fig. 4g) when compared to all leukemias that co-expressed *KMT2A-MLLT3* and an activating mutation (Supplementary Fig. 8f). Consistent with this, GSEA revealed enrichment of intracellular signaling pathways including different MAPK pathways, both at gene- and protein level (Fig. 4h, Supplementary Fig. 8i-k and Supplementary Data 7–14, 24–26). This suggests that *KMT2A-MLLT3* itself may cause sustained transcription and subsequent translation of genes involved in intracellular signaling.

**MIF promotes survival of *KMT2A-MLLT3* leukemia cells.** Mice expressing subclonal *FLT3*[N676K] succumb earlier to disease than those lacking such a mutation (Fig. 2b), raising the possibility that cells with an activating mutation, whether present at high or low MAF, provide factors that positively influence the growth or survival of leukemia cells. We therefore investigated the expression of 137 genes encoding cytokines or growth factors (Supplementary Data 29), revealing a differential expression of several

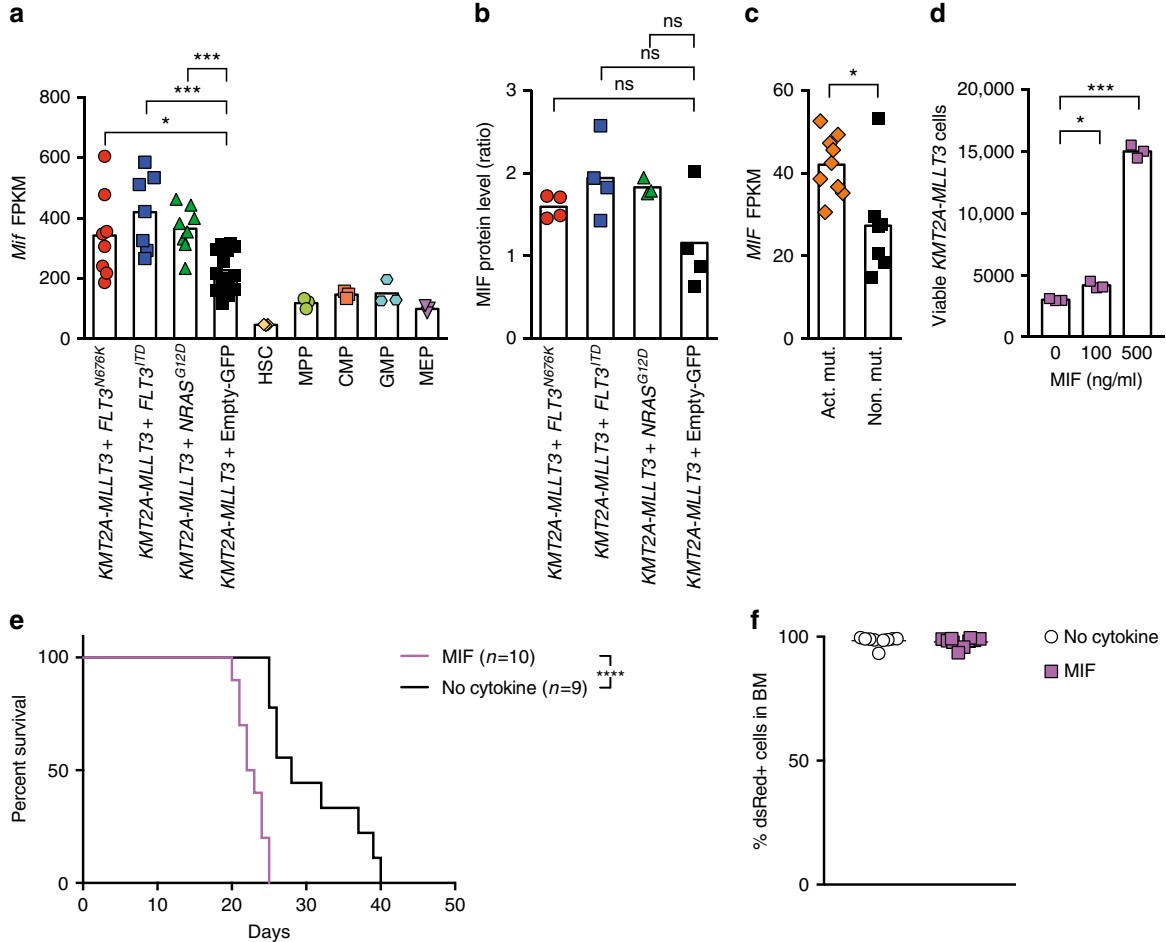

**Fig. 5** MIF promotes survival of *KMT2A-MLLT3* leukemia cells. **a** *Mif* expression (FPKM) in mouse *KMT2A-MLLT3* leukemias with or without *FLT3^ITD*, *FLT3^N676K*, or *NRAS^G12D*, as well as in sorted normal mouse hematopoietic progenitors[36] of hematopoietic stem cells (HSC), multipotent progenitors (MPP), megakaryocyte erythroid progenitor (MEP), common myeloid progenitor (CMP), and granulocyte macrophage progenitors (GMP) (Supplementary Table 4). **b** Relative MIF protein abundance in mouse *KMT2A-MLLT3* leukemic cells with or without *FLT3^ITD*, *FLT3^N676K*, or *NRAS^G12D* as determined by quantitative mass spectrometry. **c** *MIF* expression in infant *KMT2A-AFF1* ALL patients with activating mutations (dominant or subclonal) to those lacking such mutations[4]. **d** Number of viable *KMT2A-MLLT3* leukemia cells when cultured with increasing concentrations of MIF after 6 days of ex vivo culture without IL3 (n = 3). **e** Kaplan–Meier survival curve for recipient mice transplanted with serially propagated dsRed⁺ *KMT2A-MLLT3* cells[44] cultured ex vivo for 3 days with or without 500 ng/ml MIF. **f** Flow cytometric analysis on bone marrow (BM) from moribund mice revealed a majority of leukemia (dsRed⁺) cells in all mice. * $P \leq 0.05$, ** $P \leq 0.01$, *** $P \leq 0.001$, **** $P \leq 0.0001$; ns, not significant

of these genes between leukemia cells with or without an activating mutation (Supplementary Fig. 10a). One cytokine commonly upregulated in *KMT2A-MLLT3* leukemias co-expressing an activating mutation (*FLT3^ITD*, *FLT3^N676K*, or *NRAS^G12D*) as compared to those induced by *KMT2A-MLLT3* alone, and normal healthy hematopoietic subpopulations, was the macrophage migration inhibitory factor (*Mif*), a lymphokine that stimulate pro-inflammatory processes (Fig. 5a and Supplementary Fig. 10b). Proteomic analyses confirmed a higher expression of MIF in a majority of *KMT2A-MLLT3* leukemias with an activating mutation (Fig. 5b and Supplementary Fig. 11a). To establish the role of MIF in human *KMT2A*-R leukemia, we determined its expression in infant *KMT2A-AFF1*-rearranged ALL, showing that *MIF* was significantly higher expressed in patients with activating mutations (Fig. 5c)[4]. We therefore evaluated whether MIF alone was sufficient to promote growth and survival of murine *KMT2A-MLLT3* cells, which are dependent on IL3 for their survival ex vivo[10]. We found that MIF positively influenced survival of *KMT2A-MLLT3* leukemia cells when cultured without IL3 (Fig. 5d). Given that MIF influenced the survival of *KMT2A-MLLT3* leukemia cells ex vivo (Fig. 5d), we next

assessed whether MIF stimulation maintained leukemia-initiating cells (LICs) ex vivo using a transplantation assay. For this purpose, 5000 serially transplanted and LIC-enriched *KMT2A-MLLT3* c-Kit⁺ leukemia cells[44] were cultured ex vivo for 3 days with or without MIF before transplantation into sublethally irradiated recipients (Supplementary Fig. 11b). Mice transplanted with MIF treated cells had significantly shorter disease latency as compared to no cytokine control mice (22.5 days versus 28 days, $P \leq 0.0001$. Mantel–Cox log-rank test), suggesting that MIF promotes survival of LICs (Fig. 5e, f and Supplementary Fig. 11d–f).

## Discussion

We lack a full understanding of the biological processes controlling and shaping the selection of evolving leukemia clones. Mouse models have proven powerful tools to uncover genetic interactions, and the spontaneous acquisition of mutations in genes that in humans promote leukemogenesis identified herein, highlights their remarkable capacity to recapitulate human disease. Our study demonstrates that an activating mutation present only in a small proportion of *KMT2A–MLLT3* leukemia cells can

influence the rate of leukemogenesis and shorten time to leukemia onset. These data support a model whereby subclonal activating mutations are not bystanders, but influence leukemogenesis. This is the first time, to the best of our knowledge, that the functional importance of a defined leukemia subclone has been studied in the mouse. In addition, our data show that $FLT3^{N676K}$, the most frequent $FLT3$ mutation identified in our previous analysis of the mutational landscape of infant and childhood $KMT2A$-R leukemia[4], is a cooperating lesion in this disease.

The observation that subclonal activating mutations accelerate disease onset in the mouse, is consistent with our previous data showing that activating mutations, irrespective of their MAFs, associated with an average younger age at diagnosis in infants with $KMT2A$-$AFF1$ ALL[4]. Typically, subclonal $FLT3^{N676K}$ cells had a continued proliferative advantage upon secondary transplantation. However, sometimes these cells failed to expand, and we identified spontaneous de novo signaling mutations in $Ptpn11^{E69/S506}$, $Braf^{V637E}$, $Cbl^{A308T}$, and $Kras^{G12D}$, in cells bearing only $KMT2A$-$MLLT3$, thereby shifting clonal balance. Notably, further clonal evolution at the $Kras$ locus was evident upon secondary transplantation, consistent with duplication of the mutated allele and a strong selective advantage of additional $Kras^{G12D}$[45] (Fig. 3e). In human disease, data from large scale sequencing efforts have shown that patients may have more than one signaling mutation often present at different MAFs indicative of multiple leukemia clones[4,7]. Our data are consistent with different activating mutations providing different fitness to evolving leukemia cells, thus influencing their selection potential. The lack of identified spontaneous mutations in cells co-expressing $KMT2A$–$MLLT3$ and $FLT3^{ITD}$, $FLT3^{N676K}$, or $NRAS^{G12D}$ suggest that in the presence of a strong signaling mutation, the selective pressure for additional mutations within the pathway is not as prominent, at least not when the mutation is retrovirally overexpressed.

The fact that a subclonal activating mutation reduce leukemia latency raises the possibility that they secrete stimulating factors that influence other leukemia cells, as has been shown in solid tumor models with activating mutations[46–51]. MIF, a pro-inflammatory cytokine, was highly expressed in $KMT2A$-$MLLT3$ leukemia with activating mutations both in mice and in primary infant ALL patients (Fig. 5c). MIF treatment ex vivo supported the survival of mouse $KMT2A$-$MLLT3$ LICs. Thus, MIF is likely one of many factors that mediate pro-leukemic signals in vivo. MIF can inactivate p53 and thereby suppress apoptosis[52,53] and can also communicate with stromal cells, as recently described for primary AML cells, resulting in increased tumor cell survival[54]. In addition, knockout of $Mif$ delayed development of chronic lymphocytic leukemia (CLL) in mice and, similar to our data, MIF extended survival of CLL cells in vitro[55]. It is well described that genetic aberrations shape GEPs in leukemia. Mutations in $NRAS^{G12D}$, $FLT3^{ITD}$ and $FLT3^{N676K}$ further enforced $Myc$- and $Myb$-driven signatures previously associated with $KMT2A$-R leukemia[34,35,38]. MYC levels are influenced by RAS and kinase signaling[39,56], in line with activating mutations extending this transcriptional program. Signaling mutations affecting the ERK pathway, in mice as well as in infant ALL, resulted in increased transcriptional output of the pathway including negative feedback regulators, suggesting that mutated cells are insusceptible to negative regulation (Supplementary Fig. 9k)[43,45]. Interestingly, $KMT2A$-$MLLT3$ leukemias lacking a constitutively expressed signaling mutation, sustain or upregulate intracellular signaling pathways, such as RAC1 and MAPK, potentially as a contributing oncogenic event. Elevated RAC1 protein levels confer anti-apoptotic capacity in $KMT2A$-$MLLT3$ leukemia[57–59].

Taken together, these data provide a unique look into the mechanisms by which activating mutations affect leukemogenesis, also when present as a subclone. The data suggest that subclonal activating mutations, here demonstrated by $FLT3^{N676K}$, can influence leukemogenesis, possibly by providing pro-leukemic factors. In the absence of a constitutively active mutation, $KMT2A$-R leukemia cells may acquire spontaneous activating mutations, consistent with the importance of kinase/PI3K/RAS pathways as a cooperating event in this disease. The acquired mutations typically conferred cells with a competitive advantage, altered clonal fitness, and favored clonal outgrowth. Our results provide insight into the mechanisms by which signaling mutations promote leukemogenesis and it will be important to determine whether interfering with intracellular signal transduction in human $KMT2A$-R leukemia, even in the absence of an activating mutation, may be a fruitful therapeutic approach[7,60].

## Methods

**Vectors and virus production.** All vectors were based on the Murine Stem Cell Virus (MSCV) backbone and the following vectors were used: MSCV-$KMT2A$-$MLLT3$-IRES-mCherry, MSCV-$FLT3^{N676K}$-IRES-GFP, MSCV-$FLT3^{ITD}$-IRES-GFP, MSCV-$FLT3^{D835Y}$-IRES-GFP, MSCV-$FLT3^{WT}$-IRES-GFP, MSCV-$NRAS^{G12D}$-IRES-eGFP, and MSCV-$NRAS^{WT}$-IRES-eGFP. ORF cDNA clones for $Cbl$ (NM_007619) and $Ptpn11$ (NM_011202) subcloned into MSCV-eGFP were purchased from Genscript (Piscataway, NJ, USA). Site-directed mutagenesis to obtain $Cbl^{A308T}$ and $Ptpn11^{S506W}$ was performed using the QuikChange II XL Site Directed Mutagenesis Kit (Agilent Technologies, Santa Clara, CA, USA) according to the manufacturer´s instructions using the following primers; $Cbl^{A308T}$-F 5′-TGTTCCCATCGGTAGTAACATACCCAATAGCCCACT-3′, $Cbl^{A308T}$-R 5′-AGTGGGCTATTGGGTATGTTACTACCGATGGGAACA-3′, $Ptpn11^{S506W}$-F 5′-ACCATCCCCCACCTCTGGGACCGC-3′ and $Ptpn11^{S506W}$-R 5′-GCGGTCCCA-GAGGTGGGGGATGGT-3′; mutations were verified by Sanger sequencing. Empty vectors lacking the transgenes were used as controls. Retroviral supernatants were produced according to standard protocols using transient transfection of HEK 293T cells and viral containing medium was harvested 36 h later and stored at −80 °C.

**Retroviral transduction.** Whole bone marrow (BM) cells were harvested from 8–12 weeks old C57Bl/6xB6SJL mice and CD117$^+$ cells were enriched by magnetic activated cell sorting (Miltenyi Biotech, Bergisch Gladbach, Germany). Cells were pre-stimulated for 48 h in StemSpan™ SFEM (Stemcell Technologies, Grenoble, France) supplemented with 100 units/mL Penicillin and 100 g/ml Streptomycin (Thermo Scientific, South Logan, UT, USA), 50 ng/ml mSCF, 50 ng/ml hIL6, 50 ng/ml hTPO, and 20 ng/ml mIL3 (all cytokines were purchased from Peprotech, Rocky Hill, NJ, USA). Retroviral co-transduction was performed by loading a defined mixture of retroviral supernatant on Retronectin (Takara, Otsu, Japan) pre-coated wells by centrifugation for 60 min at 1000×g in 4 °C. Cells were then added, spun down at 320×g for 5 min and incubated at 37 °C for 15–18 h before bone marrow transplantation. A small aliquot of cells was saved in culture to determine the frequency of single and co-transduced cells by flow cytometry 48 h post transduction. IL3-dependent Ba/F3 cells (DSMZ, Braunschweig, Germany) were transduced in standard culture medium.

**In vivo bone marrow transplantation models.** C57Bl/6NTac, C57Bl/6xB6SJL and B6SJL mice were used throughout the experiments. C57Bl/6xB6SJL and B6SJL mice were bred in-house and sex and age matched C57Bl/6NTac were purchased from Taconic (Ejby, Denmark). For primary bone marrow transplantations, 8–12 weeks old and age-matched female C57Bl/6NTac mice were lethally irradiated with 900 cGy 18–20 h prior to transplantation and subsequently transplanted with $1 \times 10^6$ unfractionated co-transduced C57Bl/6xB6SJL BM cells. For all primary transplantations, 400,000 freshly isolated B6SJL whole BM cells were given as support, and recipient mice were given Ciprofloxacin (KRKA, Stockholm, Sweden) for 2 weeks post-irradiation in the drinking water. For secondary transplantations, 8–12 weeks old and age matched female C57Bl/6NTac mice were sublethally irradiated with 450 cGy 6 h prior to transplantation and were subsequently transplanted with $1 \times 10^6$ whole spleen cells from primary recipients. For all mice, when moribund, a PB sample was taken from vena saphena, and mice were then sacrificed by cervical dislocation. Upon sacrifice; one femur and a piece of the liver and spleen were saved for pathological evaluation. Remaining bones and spleen were made into single-cell suspension by manual trituration and BM and spleen cells were viably frozen. In addition, $1 \times 10^6$ BM cells were lysed in RLT buffer (Qiagen, Hilden, Germany) supplemented with 1% β-mercaptoethanol (Sigma-Aldrich, St. Louis, MO, USA) and stored in −80 °C before further DNA/RNA extraction. Complete blood cell count was performed on PB using Micros 60 CS

(ABX diagnostics, Montpellier, France). All mice were bred and maintained in accordance with Lund University's ethical regulations and approved by the Swedish Board of Agriculture, Malmö/Lund animal ethics committee in Lund, Sweden.

**Histopathology**. For immunohistochemistry, tissues were fixated in 4% formaldehyde (Merck, Darmstadt, Germany) and bones were subsequently decalcified using formic acid (Histolab, Göteborg, Sweden) before paraffin embedment. For structural and morphology visualization, tissue sections were deparaffinized using xylene (WVR chemicals, Fontenay-sous-Bois, France) and decreasing concentrations of ethanol (CCS Healthcare, Borlänge, Sweden) and stained using Harris hematoxylin (Histolab) and erythrosine B (Merck) before pathological evaluation. Images were taken with a Nikon BX51 microscope (Nikon Instruments Inc., Melville, NY, USA).

**Flow cytometry, cell cycle analysis, and cell sorting**. Before flow cytometric analysis of primary mouse tissues (i.e., BM and spleen) red blood cells were lysed with ammonium chloride (Stemcell technologies). Dead cells were excluded with Draq7 (1:1600, Biostatus, Shepshed, United Kingdom) or Fixable Viability Dye eFluor® 780 (1:1000, eBioscience, San Diego, CA, USA). For cell cycle analysis, cells were fixated and permeabilized using 1.6% formaldehyde (Merck) and 95% ethanol (CCS Healthcare AB) and stained with Ki-67 (1:40, clone REA183) (Miltenyi) and DAPI (1:50, Biolegend, San Diego, CA, USA). For immunophenotyping, the following antibodies were used: Ly-6G/Ly-6C/Gr-1 (1:800, clone RB6-8C5), CD11b (1:800, clone M1/70), CD45RA (1:800, clone RA3-6B2), CD3ε (1:800, clone 145-2C11), CD16/32 (1:200, clone 92), CD34 (1:20, clone MEC14.7), CD135 (1:200, clone A2F10), Ly-6A/Ly-6E/Sca-1 (1:100, clone D7), CD117 (1:100, clone ACK2), CD127/IL-7R (1:200, clone A7R34) and a lineage cocktail (1:50, CD3, clone 17A2; Ly-6G/Ly-6GC/Gr-1, clone RB6-8C5; CD11b, clone M1/70; CD45RA/B220, clone RA3-6B2; Ter-119, clone Ter-119), all from Biolegend. Flow cytometric analysis and fluorescence-activating cell sorting were performed on FACS Aria IIu or FACS Aria Fusion (BD, Franklin Lakes, NJ, USA). Data analysis was performed using the FlowJo software (FlowJo, LLC, Ashland, OR, USA).

**Ex vivo cultures**. Establishment of mouse leukemic cell lines were done by culturing sorted GFP$^+$mCherry$^+$ (for $KMT2A$-$MLLT3$ + $FLT3^{N676K}$, $KMT2A$-$MLLT3$ + $FLT3^{N676K}$ and $KMT2A$-$MLLT3$ + $NRAS^{G12D}$) or mCherry$^+$ (for $KMT2A$-$MLLT3$ + Empty-GFP) leukemic BM cells for 1 week in "C10" media, consisting of RPMI-1640 with L-glutamine (Thermo Scientific), 10% fetal bovine serum (FBS, Thermo Scientific), 100 units/mL Penicillin and 100 g/ml Streptomycin (Thermo Scientific), 55 μM ß-mercaptoethanol (Sigma-Aldrich), 0.1 mM non-essential amino acids (Sigma-Aldrich), 1 mM sodium pyruvate (Thermo Scientific) and 10 mM HEPES (Thermo Scientific) supplemented with 100 ng/ml mSCF, 10 ng/ml hIL6 and 10 ng/ml mIL3 (Peprotech)[10], after which mSCF and hIL6 were withdrawn and mIL3 was decreased to 1 ng/ml for continuous culturing.

**Western blot analysis**. Ba/F3 cells and Ba/F3 cells transduced with either $FLT3^{WT}$-GFP, $FLT3^{N676K}$-GFP, $FLT3^{ITD}$-GFP, $FLT3^{D835Y}$-GFP, $NRAS^{WT}$-eGFP, and $NRAS^{G12D}$-eGFP sorted for GFP$^+$ were kept in RPMI-1640 with L-glutamine (Thermo Scientific), 10% FBS (Thermo Scientific), 100 units/mL Penicillin and 100 g/ml Streptomycin (Thermo Scientific) supplemented with 10 ng/ml mIL3 (Peprotech) for routine culture. Primary mouse leukemia or transduced Ba/F3 cells were washed three times with cold PBS (GE Healthcare Life Sciences, Logan, UT, USA), starved for 3 h at 37 °C in C10 medium without FBS and IL3, and then lysed in 1% Triton X-100 lysis buffer (Sigma-Aldrich) containing 1 mM sodium orthovanadate (Sigma-Aldrich), 1% Trasylol (Sigma-Aldrich) and 1 mM PMSF (Sigma-Aldrich), for 15 min on ice. Cell lysates were mixed with 2 × laminal buffer (1:1) (Bio-Rad, Hercules, CA, USA) and incubated at 95 °C for 5 min. Proteins were separated on 8.5% SDS-PAGE gels (Bio-Rad), and transferred to PVDF membranes (Merck Millipore, Burlington, MA, USA). Membranes were blocked for 1 h in 5 % non-fat dry milk (Bio-Rad) in 0.2 % PBS-Tween (Sigma-Aldrich). For western blots of MIF, primary leukemia cells were lysed in M-Per™ Mammalian Protein Extraction Reagent (Thermo scientific) containing cOmplete™ (Sigma-Aldrich) and 0.5 M EDTA (Thermo scientific) for 10 min. Cell lysates were mixed with 4× NuPage sample buffer (Thermo scientific) and 50 mM DTT (Invitrogen), and incubated at 85 °C for 7 min. Proteins were separated on 10% Bis-Tris gel (Invitrogen), and transferred to PVDF membranes (GE Healthcare Life Sciences). Membranes were incubated overnight at 4 °C with primary antibodies followed by 1 h incubation with the secondary horseradish peroxidase-conjugated antibody. Immunodetection was performed with Luminata Forte Western HRP Substrate (Merck Millipore) or Pierce™ ECL Western Blotting Substrate (Thermo scientific) and LAS-3000 CCD or LAS-4000 camera (Fujifilm, Tokyo, Japan). The following antibodies were used: Rabbit anti-phospho-ERK1/2 (pThr202/pThr204; 1:1000, sc-16982), goat anti-AKT (1:1000, sc-1619), rabbit anti-ERK2 (1:1000, sc-154), mouse anti-β-Actin (1:1000, sc-47778), mouse anti-N-RAS (1:1000, sc-31), rabbit anti-STAT5 (1:1000, sc-835), HRP-Rabbit Anti-Goat IgG (1:100000, sc-2768) (all Santa Cruz Biotechnology, Dallas, TX, USA); rabbit anti-phospho-AKT (pSer473; 1:5000, ab81283), rabbit anti-MIF (1:500, ab7207), and rabbit anti-phospho-STAT5 (pTyr694; 1:1000, ab32364) (all from Abcam, Cambridge, UK); mouse anti-phospho-P38 (pThr180/pTyr182; 1:250, #612289) and mouse anti-P38 (1:250,

#612169) (BD Transduction Laboratories, San Jose, CA, USA); HRP-Goat Anti-Rabbit IgG (1:100000, #65–6120) and HRP-Goat Anti-Mouse IgG (1:100000, #62–6520) (both from Thermo Scientific); HRP-Sheep Anti-Mouse IgG (1:5000, NA931) and HRP-Donkey Anti-Rabbit IgG (1:5000, NA9340) (both from GE Healthcare Life Sciences); 4G10 (1 μg/ml, #05–1050; Merck Millipore); anti-FLT3 (1 μg/ml) homemade and previously described[61].

**DNA PCR validation of $FLT3^{N676K}$ subclones**. PCR was performed on DNA extracted from BM for samples SJ017014 (minor clone; 0.77% $FLT3^{N676K}$), SJ018185 (minor clone; 0.38% $FLT3^{N676K}$), SJ046281 (positive control; 82.7% $FLT3^{N676K}$) and SJ047106 (negative control), using the AccuPrime™ Taq DNA Polymerase System (Life Technologies) according to the manufacturer's instructions (94 °C, 2 min; and 35 cycles of: 94 °C, 0.5 min; 62 °C, 0.5 min; 68 °C, 1 min). Primers were designed using Primer-BLAST (https://www.ncbi.nlm.nih.gov/tools/primer-blast/) and the following primers were used: FLT3-5′-AAAATG-GATGGCCCCCGAAA-3′ and IRES-5′- GAGAGGGGCGGAATTGATCC-3′. PCR products were analyzed by gel electrophoresis. PCR reactions were purified using ExoSAP-IT (Affymetrix, Santa Clara, CA, USA) and sent to GATC Biotech (Constance, Germany) for Sanger sequencing using above primers.

**Targeted amplicon deep sequencing and analysis**. A custom targeted panel of 41 genes (Supplementary Table 1) was constructed based on two unbiased sequencing efforts of infant, childhood and adult $KMT2A$-R leukemia[4,7], and selected genes were either recurrently mutated in its respective study or part of a PI3K/RAS/kinase- or epigenetic pathway. An additional nine genes were manually added (Supplementary Table 1). PCR amplicon library preparation was performed using Illumina TruSeq Custom Amplicon Low Input according to manufacturer´s instructions (v03). In short, 20 ng of mouse genomic DNA was hybridized with the custom oligo pool, bound oligos were extended and ligated after which the libraries were amplified. Library concentrations were determined by the Qubit Fluorometer (Life Technologies), libraries were pooled and 2 × 150 bp paired-end sequencing was performed using the Illumina NextSeq 500 (Illumina). Paired-end reads from the targeted amplicon deep sequencing were aligned to the mouse genome mm9 on BaseSpace (Illumina) using a banded Smith–Waterman algorithm and Bedtools was used to calculate coverage. Mpileup files were created using SAMtools (1.3.1)[62] and variant calling was performed using VarScan (2.4.1)[63]. Putative mutations were filtered based on quality and resultant variants were manually reviewed using Bambino[64]. For calculations of mutant allele frequencies, the number of reads required for the mutant allele was ≥100, otherwise data from the PCR amplicon validation was used. Fish plots depicting clonal evolution between primary and secondary recipients were based on mutant allele frequencies for single nucleotide variants as determined by the targeted amplicon sequencing (Supplementary Data 2) and frequencies of cells displaying aberrant interphase FISH analysis signals (Supplementary Table 3).

**Validation of somatic mutations**. Validation of each mutation was performed in matched primary, secondary recipient, and constitutional DNA, using PCR amplicon deep sequencing. Primers were designed using Primer-BLAST (sequences available upon request). PCR was performed using Platinum Taq DNA Polymerase kit (Life Technologies) according to the manufacturer's instructions and products were analyzed using gel electrophoresis. PCR amplicons were purified using AMPure XP beads (Beckman Coulter Inc., Brea, CA) and prepared for sequencing using the Nextera XT DNA Sample Preparation Kit and Index Kit (Illumina). 2 × 150 bp paired-end sequencing was performed using the Illumina NextSeq 500 (Illumina) and paired-end reads were aligned to mm9 using BWA (0.7.12).

**Fluorescence in situ hybridization (FISH)**. FISH was performed according to standard methods on cultured BM cells from the primary (SJ016338) and secondary (SJ046295) recipients with acquired $Kras^{G12D}$, utilizing BACs BMQ-415F20, BMQ-422H6 and BMQ-189B18 (Source BioScience, Nottingham, UK), overlapping the Kras locus, and a whole chromosome paint probe for chromosome 6 (MetaSystems Probes GmbH, Altlussheim, Germany). A minimum of 100 interphase and 10 metaphase cells were scored for each sample.

**Structural analyses**. The three-dimensional structure of the N-terminal domain of human CBL (PDB ID: 2Y1M) was aligned with the complex structures of human CBL/ZAP-70 (PDBID: 2Y1N), CBL/SYK (PDBID: 3BUW) and CBL/EGFR (PDBID: 3BUO) applying the align command in the PyMOL Molecular Graphics System, Version 1.8 Schrödinger, LLC. Only one CBL representation is shown for clarity.

**Ba/F3 cell culture and cytokine independence assay**. To analyze cytokine independence, Ba/F3 cells were transduced with the retroviral constructs $Cbl^{WT}$-eGFP + $FLT3^{WT}$-mCherry, $Cbl^{A308T}$-eGFP + $FLT3^{WT}$-mCherry or $FLT3^{WT}$-mCherry, viable cells were sorted based on expression of GFP$^+$mCherry$^+$ (for $Cbl^{WT}$-eGFP + $FLT3^{WT}$-mCherry and $Cbl^{A308T}$-eGFP + $FLT3^{WT}$-mCherry) or mCherry$^+$ (for $FLT3^{WT}$-mCherry) and expanded in RPMI-1640 with L-glutamine (Thermo Scientific), 10% FBS (Thermo Scientific), 100 units/mL Penicillin and 100

g/ml Streptomycin (Thermo Scientific) supplemented with 10 ng/ml mIL3 (Peprotech). For cytokine independency assay, cells were washed three times in PBS + 2% FBS and seeded in triplicates at a density of $0.5 \times 10^6$ cells/ml in 1 ml in 12-well plates. Cells were counted continuously using CountBright beads (Life Technologies) on a FACS Fortessa (BD) for up to 19 days. Ba/F3 cells treated with 10 ng/ml mIL3 was included as a positive and Ba/F3 without mIL3 was used as a negative control. Cells were split when cell numbers reached $2 \times 10^6$ cells/ml to 0.2 million/ml.

**DNA/RNA extraction and RNA sequencing.** RNA and DNA were extracted using AllPrep DNA/RNA Mini Kit (Qiagen) according to the manufacturer's instructions. The DNA quantity and quality was assessed by the NanoDrop 1000 Spectrophotometer (Thermo Fisher Scientific, Waltham, MA, USA) and the Qubit Fluorometer (Life Technologies, Paisley, UK), respectively. The RNA quantity and quality was assessed by the NanoDrop 1000 Spectrophotometer (Thermo Fisher Scientific) and the Agilent 2100 Bioanalyzer (Agilent Technologies), respectively. Selection of poly(A) mRNA and subsequent cDNA synthesis was performed using the Illumina TruSeq RNA sample preparation kit (Illumina, San Diego, CA, USA) according to the manufacturer's instructions, but with a modified RNA fragmentation step lowering the incubation at 94 °C from 8 min to 10 s. Quality and size of the library were analyzed using the Agilent 2100 Bioanalyzer (Agilent Technologies) and $2 \times 80$ bp paired-end sequencing was performed on Illumina NextSeq 500 (Illumina).

**Analysis of RNA sequencing data.** Paired-end reads from RNA sequencing were aligned to the mouse genome mm9 using BWA (0.5.10) aligner and a bam file was constructed from which coverage was calculated using an in-house pipeline. Transcript expression levels were estimated as fragments per kilobase of transcript per million mapped fragments (FPKM) and gene FPKM was calculated using Cuffdiff2[65] and log[2] transformed. A FPKM cutoff of ≥0.5 was used to define "expressed" genes and genes not expressed in at least one sample were excluded. For the expression matrix with normal hematopoietic populations[36] data was quantile-normalized. Visualization and statistical analysis was performed using Qlcore Omics Explorer 3.1 (Qlcore, Lund, Sweden). GSEA was performed using GSEA v2.2.0[37] with pre-ranked gene lists and gene ontology was performed using DAVID[66].

**Sample preparation for proteome analysis.** FACS was used to isolate GFP$^+$mCherry$^+$ leukemic BM cells from three $KMT2A\text{-}MLLT3 + NRAS^{G12D}$ mice (SJ017001, SJ017002 and SJ017003), four $KMT2A\text{-}MLLT3 + FLT3^{ITD}$ mice (SJ017005, SJ017006, SJ017007 and SJ017008), four $KMT2A\text{-}MLLT3 + FLT3^{N676K}$ mice (SJ017009, SJ017010, SJ017011 and SJ017012) as well as mCherry$^+$ leukemic BM cells from four $KMT2A\text{-}MLLT3 +$ Empty-GFP mice (SJ018148, SJ018150, SJ018151 and SJ018152). Cells were stored as cell pellets in −80 °C. Cell pellets corresponding to $1 \times 10^6$ FACS-sorted cells were lysed with 0.1% RapiGest (Waters, Milford, MA, USA) in 55 μl 200 mM HEPES (pH 8), heated at 90 °C for 5 min, followed by sonication for 20 min and removal of cell debris by centrifugation. Cysteine disulfide bonds were reduced with 5 mM DTT for 30 min at 56 °C, alkylated with 10 mM iodoacetamide for 30 min at room temperature in the dark, and proteins were digested with Mass Spectrometry Grade Lysyl Endopeptidase (Wako, Osaka, Japan) at enzyme:protein ratio 1:50, at 37 °C overnight. Peptides were differentially labeled with Tandem Mass Tag isobaric labeling (TMT 6-plex, Thermo Scientific) according to the manufacturer's instructions with slight modifications. Briefly, 0.8 mg TMT reagents were dissolved in 40 μl acetonitrile and 5 μl was added to each sample. After 30 min (shaking), another 5 μl TMT reagent was added and samples were shaken for 30 min. The reaction was quenched by adding 8 μl of 5% hydroxylamine and samples were incubated at RT for 15 min. RapiGest was then precipitated by addition of 10 μl 10% trifluoroacetic acid and incubation at 37 °C for 45 min. Following centrifugation, supernatants were collected and samples were desalted with SepPak $C_{18}$ cartridges (Waters) by washing the columns with acetonitrile, conditioning and loading samples in 0.1% (v/v) trifluoroacetic acid, washing with 0.1% formic acid and eluting peptides with 80% (v/v) acetonitrile/0.1% (v/v) formic acid. For the control mix, four $KMT2A\text{-}MLLT3 +$ Empty-GFP samples (all labeled with TMT 126) were mixed at this stage and split into three equal volumes.

Samples were dried by vacuum centrifugation, reconstituted in IPG rehydration buffer and fractionated according to manufacturer's instructions using pH 3–10 IPG strips and 3100 OFFGEL fractionator (Agilent Technologies). The 12 resolved fractions were acidified and desalted with $C_{18}$ Ultra-Micro Spin Columns (Harvard Apparatus, Holliston, MA, USA). Peptide samples were dried by vacuum centrifugation and stored at −20 °C until further use. Samples were reconstituted in 4% acetonitrile/0.1% formic acid prior to MS analysis.

**Proteome analysis.** MS analyses were carried out on the Orbitrap Fusion Tribrid MS system (Thermo Scientific) equipped with a Proxeon Easy-nLC 1000 (Thermo Fisher). Injected peptides were trapped on an Acclaim PepMap C18 column (3 μm particle size, 75 μm inner diameter × 20 mm length, nanoViper fitting), followed by gradient elution of peptides on an Acclaim PepMap C18 100 Å column (3 μm particle size, 75 μm inner diameter × 150 mm length, nanoViper fitting). The

mobile phases for LC separation were 0.1% (v/v) formic acid in LC-MS grade water (solvent A) and 0.1% (v/v) formic acid in acetonitrile (solvent B). Peptides were loaded with a constant flow of solvent A at 9 μl/min onto the trapping column followed by peptide elution via the analytical column at a constant flow of 600 nl/min. During the elution step, the percentage of solvent B increased in a linear fashion from 5% to 10% in 2 min, then to 25% in 85 min and finally to 60% in 20 min. The peptides were introduced into the mass spectrometer via a Stainless Steel Nano-bore emitter 150 μm OD × 30 μm ID; 40 mm length (Thermo Fisher Scientific) and a spray voltage of 2.0 kV was applied. The capillary temperature was set at 275 °C.

Data acquisition was carried out using a data-dependent SPS-MS3 method with cycle time of 3 s. The master scan was performed in the Orbitrap in the range of 380–1580 $m/z$ at a resolution of 120,000 FWHM. The filling time was set at maximum of 50 ms with limitation of $4 \times 10^5$ ions. Ion trap CID-MS2 was acquired using parallel mode, filling time 50 ms with limitation of $1.5 \times 10^4$ ions, a precursor ion isolation width of 0.7 $m/z$ and resolution of 30,000 FWHM. Normalized collision energy was set to 35%. Only multiply charged (2+ to 5+) precursor ions were selected for MS2. The dynamic exclusion list was set to 30 s and relative mass window of 5 ppm. Precursor selection range for MS3 was set to $m/z$ range 400–1200 in MS2. Orbitrap HCD-MS3 scans were acquired in parallel mode with synchronous precursor selection (ten precursors), normalized collision energy of 55%, filling time 120 ms with limitation of $1 \times 10^5$ ions and a resolution of 15,000 FWHM in a range of 100–500 $m/z$.

**Proteome data analysis.** MS raw data were processed with Proteome Discoverer (version 2.1, Thermo Scientific). Enzyme was set to LysC and a maximum of two missed cleavages were allowed. TMT-K and TMT N-term were set as static modifications. Data were annotated using SEQUEST search engine against the Uniprot mouse database (downloaded 2016.05.29) containing 79,920 proteins to which 197 frequently observed contaminants had been added, as well as the trans and reporter genes used in this study. Maximal allowed precursor mass tolerance was set to 10 ppm. 6756 proteins were identified of which 5941 fulfilled an FDR < 0.05. Of identified proteins, 94.9% could be linked to their corresponding mRNA in the RNAseq data of expressed genes (6411 out of 6756).

Statistical analysis was performed using the Limma package in R/Bioconductor[67]. After fitting a linear model to the data, an empirical Bayes moderated $t$-test was used for the protein ratios. $P$-values were adjusted for multiple testing with Benjamini and Hochberg's method. To match proteomic and transcriptomic signatures, average log2 protein ratios and average mRNA ratios (as compared to $KMT2A\text{-}MLLT3 +$ Empty-GFP) were combined for matching gene symbols and 2D enrichment was performed using Perseus (v1.5.5.3)[68] with an Benjamini-Hochberg FDR of 0.02.

**MIF ex vivo treatment assays.** For the cytokine assay, established mouse $KMT2A\text{-}MLLT3 +$ Empty-GFP leukemic cells were washed once in cold PBS (GE Healthcare Life Sciences), resuspended in C10 medium supplemented with 25 ng/ml mSCF (Peprotech) and plated in 100 μl at a density of 50,000 cells per well in a 96-well U-bottom plate and 5 μl recombinant human MIF (Peprotech) was added at a final concentration of 100 or 500 ng/ml, and PBS (GE Healthcare Life Sciences) was used as control. Cells were counted on day 6 after seeding using CountBright beads (Life Technologies, Eugene, OR, USA) on a FACS Fortessa (BD). Data analysis was performed using the FlowJo software (FlowJo, LLC).

For the ex vivo transplantation assay, serially propagated dsRed$^+$ $KMT2A\text{-}MLLT3$ leukemia cells[44] were used[69]. Briefly, 5000 freshly isolated CD117$^+$ quaternary transplant leukemia cells were treated ex vivo with or without 500 ng/ml MIF (Peprotech) in StemSpan™ SFEM (Stemcell Technologies) for 3 days before transplantation into sublethally irradiated (600 cGy) recipients. Blood sampling was performed 3 weeks post transplantation and at disease manifestation, BM cells from femurs and tibias were collected, and the percentage of dsRed$^+$ leukemia cells were determined using flow cytometry.

**Statistical analysis.** Differences between groups were assessed by unpaired or paired (when noted) Students $t$-test. Correlation was assessed using Spearman´s rank correlation coefficient. Statistical analysis of survival curves was performed using Mantel-Cox log-rank test. All graphs show mean with all individual data points. All analyses were performed with Prism software version 6.0 (GraphPad software).

**Code Availability.** GSE106714 and PXD008213. Referenced accession codes: SRP100343. Protein Data Bank: 2Y1M, 2Y1N, 3BUW, 3BUO, and 4A4B.

**Data availability.** RNA sequencing data have been deposited under the accession number GSE106714 in the Gene Expression Omnibus (GEO). Mass spectrometry proteome data have been deposited with the identifier PXD008213 to the ProteomeXchange Consortium via the Proteomics Identifications Database (PRIDE) partner repository[70]. All other relevant data are available from the corresponding author upon request.

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

## Acknowledgements

We thank Andrea Biloglav for technical support and Kevin Shannon for critical reading of the manuscript. This work was supported by The Swedish Childhood Cancer Foundation, The Swedish Cancer Society, The Swedish Research Council, The Knut and Alice Wallenberg Foundation, BioCARE, The Gunnar Nilsson Cancer Foundation, Ellen Bachrachs Memorial Foundation, The Craaford Foundation, The Per-Eric and Ulla Schyberg Foundation, The Nilsson-Ehle Donations, The Wiberg Foundation, Governmental Funding of Clinical Research within the National Health Service. Support from the Swedish National Infrastructure for Biological Mass Spectrometry (BioMS) is gratefully acknowledged. A.H. is partly funded by The Georg Danielsson Foundation for Hematopoietic Disease.

## Author contributions

A.K.H.A. designed the study; A.H.W. and A.K.H.A. designed experiments; A.H.W., H.S., J.H., J.U.K., J.L., R.R., C.G.R, and K.P. performed experiments; M.P., M.P.W., G.S., P.G., J.Z., and J.M performed computational data analyses; A.H.W and J.H. analyzed mass spectrometry data; S.N. and T.A.G. provided sequencing data of normal mouse progenitors; A.H.W., M.P., M.P.W., J.M., and A.K.H.A. analyzed sequencing data; K.L.P performed structural visualization; A.H. performed pathological evaluation; L.R., J.R.D., K.P., M.J., and T.A.G. performed critical reading of the manuscript; A.H.W. and A.K.H.A. wrote the manuscript.

## Additional information

**Competing interests:** The authors declare no competing interests.

