## [Peer Review File · Nature Communications]

Reviewers' comments:

Reviewer #1 (Remarks to the Author):

The manuscript by Hyrenius-Wittsten et al. explores the activating mutations in KMT2A rearrangement leukemia that drives clonal expansion and accelerates the disease. The authors co-transduced retroviral vectors expressing KMT2A-MLLT3 and FLT3-ITD-GFP, FLT3N676K -GFP, NRASG12D-GFP into mouse c-Kit+ hematopoietic stem and progenitor cells. They show that these activating mutations accelerate AML as compared to the KMT2A-MLLT3 alone. The authors then focused on the FLT3N676K mutation KMT2A-MLLT3 leukemia and performed serial transplants. They found that this activating mutant is more aggressive in secondary recipients, driving the leukemia onset and reducing the disease latency. Furthermore, they reported de novo activating mutations in Cbl, Braf, Kras and Ptpn11 emerging from the KMT2A-MLLT3 alone expressing cells and showed that these give the leukemia clonal advantage. The authors characterized the gene expression signatures in these KMT2A-MLLT3 leukemia containing these acquired mutations.

Overall this manuscript contributes to our understanding of the complexity of AML by demonstrating the fitness potential of competing clonal populations in leukemia and how de novo mutations can also outcompete other driver mutations.

Several concerns need to be address to strengthen the study:

Major concerns:

1. Despite the characterization of the clonal populations and gene expression profiling the manuscript is somewhat convoluted by the various analysis which converges on MEK/ERK, MYB/MYC and then MIF. The link between everything should be strengthened and better described. One experiment that would greatly strengthen the paper would be to overexpress MIF in KMT2A-MLLT3 alone expressing cells as a subclone and to determine if this can accelerate the leukemia in primary transplants.
2. The authors should describe why mice with reduced latency also have smaller spleens and less disease burden (Supplemental Fig 1 e-h and supplemental Fig 2).
3. The authors claimed that secondary malignancies of the activating mutations have reduced disease latency, however it doesn't seem true comparing Fig 1a and Fig 1d. If anything, it is the KMT1A-MLLT3 alone accelerating the leukemia the most. A statistical test comparing the primary to each mutation's secondary should be done or the authors should state that the leukemias are not accelerated.
4. What do authors want to conclude by showing smaller fraction of cells co-expressing FLT3N676K and KMT2A-MLLT3 in SPL compared to in BM in primary recipients? Fig 2c. why did the authors not show %GFP+ in the same organ in primary and secondary transplantation?
5. The section about MIF has several issues. To which data/figure authors refer when they stated mice expressing subclonal FLT3N676K succumb earlier to the disease than other mice lacking this mutation? At the beginning of this section, authors tried to explain the advantages of subclonal expressing FLT3N676K compared to those lacking this mutation, but later on they went on to show that Mif is higher in all leukemias, including the ones co-expressing other activating mutations (FLT3-ITD and NRASG12D)., Fig 5c could have been strengthened with a western blot.

Minor concerns:

1. Supplemental Fig 1c should be moved to the BaF3 experiment or after the gene expression profiling.
2. The authors should include a normal control for the level of p-p38 in supplemental Fig 3 to say it is high in all leukemias.
3. Page 6, Supplemental Fig 5a, which experiment corresponds to which ratio of KMT2A-MLLT3+ FLT3N676K : KMT2A-MLLT3 alone. It is confusing how the data are presented. It does not look like it is the ratio of cells in Supplemental Fig 5a. There should be a schematic explanation.

4. Authors should provide more detail for how the fish plots were determined in the methods.
5. In Fig S5h, the axis should be %GFP+mCherry+

Reviewer #2 (Remarks to the Author):

In leukemias associated with MLL translocations, concomitant mutations in KRAS, NRAS and FLT3, among others, occur frequently. The current study follows on from Dr Andersson's prior work (Andersson et al., 2015, Nat. Genetics) which characterised the mutational landscape of infant ALL.

Here, the authors show (as is well established – see refs 13-20 in the manuscript) that coexpression of signal transduction activating mutations in FLT3/RAS with MLL-AF9 accelerate onset of AML in a mouse model.

In cell mixing transplant experiments (with MLL-AF9 cells), the authors show that MLL-AF9/FLT3 N676K cells (which cycle more rapidly than MLL-AF9 only control cells) contribute to reduced disease latency when either a major clone or a sub-clone. Mice secondarily transplanted with leukemic splenocytes from mice with higher proportions of MLL-AF9/FLT3 N676K cells exhibited shorter disease latencies; the size of MLL-AF9/FLT3 N676K clones mostly expand in secondary recipients versus MLL-AF9 only control clones.

Targeted sequencing of (?) 122 murine leukemias was performed & point mutations in Braf, Cbl, Kras and Ptpn11 were identified in a minority of cases – mutant allele frequency analysis demonstrated some clones to be major & others to be minor sub-clones.

An acquired CBL A308T mutation in MLL-AF9 AML cells is argued to provide the clone with the ability to "out compete" an MLL-AF9/FLT3 N676K sub-clone, although the equivalent mutation has not been reported in human malignancy. Functional studies in BAF/3 cells are shown, although these are difficult to interpret.

Coexpression of these acquired mutations (although CBL A308T not done) with MLL-AF9 shortens leukemia latency.

RNA sequencing was performed on many of these leukemias (and normal progenitor cells as comparators) and, perhaps unsurprisingly given they cycle faster, leukaemias with co-expressed activating mutations differentially express specific gene sets such as MYC targets, ribosomal genes and translation genes.

Mass spectrometry/proteomics was also performed, although the results of these analyses do not seem to be discussed in detail (e.g. where is the list of identified proteins?) and may perhaps more usefully be omitted.

There follows extensive gene set enrichment analysis, some of which is difficult to understand.

Finally, the authors note that MLL-AF9/FLT3 N676K AML cells express higher levels of macrophage migration inhibitory factor (Mif) transcripts by comparison with control MLL-AF9 AML cells. They also state that protein levels are higher (line 312) but the accompanying Figure 5B shows no significant difference. An experiment is shown which implies that Mif might be able to sustain survival of growth factor deprived AML cells in vitro, although the panel (Figure 5E) & legend do not make it easy to understand precisely what experiment has been performed. No in vivo data are provided. The argument that a minor MLL-AF9/FLT3 N676K AML clone shortens AML latency by secreting factors that alter the behaviour of the major clone is not well supported by the data here; it seems more probable that it shortens latency by contributing to the total number of leukaemia cells in the mouse.

This is clearly a huge amount of work; the manuscript is generally well written and the figures well presented.

It is of interest that mutations frequently found in human MLL leukaemias are spontaneously acquired during the course of leukaemia development in mouse models of the same. It is noteworthy that, as in humans, their role seems to be to accelerate growth of the leukemic clone rather than being absolutely required for transformation or development of full-fledged leukemia.

There are some significant concerns. First, it is hard to discern what is truly novel here: the association of activating mutations in signal transduction pathways with MLL leukemias is well established, likewise the fact that the introduction of these mutations accelerates (through enhancing proliferation) the onset of leukemia. The RNAseq, proteomic and GSEA analyses reveal the results one would expect if two MLL-leukemia clones are compared, one of which is cycling faster than the other. Overall the conclusions presented here seem incremental.

An additional point is the extent to which the experimental system models human disease. Seemingly in the previous work (Andersson et al., 2015, Nat. Genetics) there were no FLT3 mutations associated with MLL-AF9 (Figure 2b). Instead the two FLT3 N676K mutations were found as major clones in MLL-AF4 cases (Table S20). The biology of MLL-AF4 versus MLL-AF9 is quite distinct, as evidenced by the great difficulty there has been over the years modelling MLL-AF4 infant ALL (in contrast to MLL-AF9 which readily transforms BM stem and progenitor cells). Might FLT3 N676K have a role specifically in infant MLL-AF4 disease?

Other comments

1. Lines 119-121 & accompanying Figure S5A; the precise meaning of the Figure in relation to the text is unclear. How does the figure illustrate the experiment described in the text? Please make this clearer.
2. Lines 123-125; not clear to which of the three experiments mentioned on line 119 the authors refer to here. Likewise for Figure S5c – which graph refers to 1:28, 1:41 and 1:156 cell mixing experiment?
3. Lines 167-170; not clear which murine leukemias exhibited which mutations. Please make this clear. Please state whether these mutations confer constitutive signalling activity.
4. Figure S7A; this is not clear – does this graph show growth of BAF/3 cells without IL3? If not, why do the control cells seemingly die? Also, why include FLT3 transduction in this experiment? If the CBL A308T mutation is truly able to replicate constitutively active RAS pathway signalling, surely it should read out on its own in this assay?
5. Figure 3g; this is not clear – were there three separate cohorts of mice, each with 3 mice? Please make this clear.
6. Please show the gene names that permit the cluster analysis shown in Figure 4A.
7. Lines 285-6; it is not clear what is meant by “known output genes” and “negative feedback regulators...”. Please clarify. This makes the conclusions presented in lines 285-293 difficult to follow or understand.
8. Line 294/295 – what is the comparator population here? Please make this clear. It is currently difficult to understand the conclusion here.
9. Gene names in Figure S10A are too small to read. Please enlarge.
10. The discussion largely repeats the findings from the results section and could be substantially shortened.

Response to Reviewers Comments and Questions

NCOMMS-17-11808-T, “De novo activating mutations drive clonal evolution and enhance clonal fitness in *KMT2A*-rearranged leukemia.”

Reviewer #1 (Remarks to the Author):

The manuscript by Hyrenius-Wittsten et al. explores the activating mutations in *KMT2A* rearrangement leukemia that drives clonal expansion and accelerates the disease. The authors co-transduced retroviral vectors expressing *KMT2A-MLLT3* and *FLT3-ITD-GFP*, *FLT3N676K -GFP*, *NRASG12D-GFP* into mouse c-Kit⁺ hematopoietic stem and progenitor cells. They show that these activating mutations accelerate AML as compared to the *KMT2A-MLLT3* alone. The authors then focused on the *FLT3N676K* mutation *KMT2A-MLLT3* leukemia and performed serial transplants. They found that this activating mutant is more aggressive in secondary recipients, driving the leukemia onset and reducing the disease latency. Furthermore, they reported de novo activating mutations in *Cbl*, *Braf*, *Kras* and *Ptpn11* emerging from the *KMT2A-MLLT3* alone expressing cells and showed that these give the leukemia clonal advantage. The authors characterized the gene expression signatures in these *KMT2A-MLLT3* leukemia containing these acquired mutations.

Overall this manuscript contributes to our understanding of the complexity of AML by demonstrating the fitness potential of competing clonal populations in leukemia and how de novo mutations can also outcompete other driver mutations. Several concerns need to be address to strengthen the study:

Reviewer #1: Major Comments/Questions

1. Despite the characterization of the clonal populations and gene expression profiling the manuscript is somewhat convoluted by the various analysis which converges on MEK/ERK, MYB/MYC and then MIF. The link between everything should be strengthened and better described. One experiment that would greatly strengthen the paper would be to overexpress MIF in *KMT2A-MLLT3* alone expressing cells as a subclone and to determine if this can accelerate the leukemia in primary transplants.

Response: We have modified the gene expression sections on pages 13-14 to improve the link between the analyses performed and hope that the rationale now is clearer.

To further address the role of Mif in leukemogenesis, we have performed two different *in vivo* experimental approaches. In the first approach, we tried to retrovirally overexpress Mif together with *KMT2A-MLLT3* to determine its effect on overall survival. Overexpression of Mif in leukemic cells did not translate to a significant difference in survival of recipient mice, possibly due to insufficient levels of Mif *in vivo*. In line with this, western blots on primary leukemias from these experiments showed no apparent difference in Mif levels between leukemic mice “overexpressing” Mif and control mice (*KMT2A-MLLT3* + Empty vector). Thus, even though control experiments in BaF3 cells showed that the construct expresses Mif, it does not produce high levels of MIF *in vivo*. Since we were unable to confirm overexpression of Mif in these transplantations we feel that we are unable to draw decisive conclusions from this experiment and have therefore decided not to include this data in the revised version of the manuscript.

In the second approach, we determined if MIF had the ability to maintain leukemia initiating cells (LICs) using a well-established model of enriched dsRed⁺ *KMT2A-MLLT3* LICs that was established by Benjamin Ebert (Miller, P. G. et al., *Cancer Cell*, 2013). In this model,

GMPs transduced with *KMT2A-MLLT3* were serially transplanted to enrich for LICs. These cells were then treated *ex vivo* with or without MIF for 3 days and transplanted into sublethally irradiated recipients. This showed that MIF preserved the number of *KMT2A-MLLT3* LICs and that this translated to a significantly shorter survival (Figure 5e-f and Supplementary Figure 11b-f). These data have been added to the manuscript in the result section page 15, lines 343-351 and in Figure 5e-f and Supplementary Figure 11b-f.

2. The authors should describe why mice with reduced latency also have smaller spleens and less disease burden (Supplemental Fig 1 e-h and supplemental Fig 2).

Response: Our data is similar to that observed for *KMT2A-MLLT3+FLT3-ITD* in the study by Armstrong and colleagues (Stubbs M. C. et al., *Leukemia*, 2008) in which mice with *KMT2A-MLLT3+FLT3-ITD* succumb earlier to disease, but with less disease burden as compared to mice receiving only *KMT2A-MLLT3*. With regard to disease burden, we believe that there is a relationship between disease latency and splenomegaly/leukocytosis. The longer the disease process, the more pronounced are the disease parameters. To visualize this, we plotted disease latency versus both splenomegaly and leukocytosis for *KMT2A-MLLT3+Empty-GFP*, which both revealed a significant correlation (see attached figures

below).

3. The authors claimed that secondary malignancies of the activating mutations have reduced disease latency, however it doesn't seem true comparing Fig 1a and Fig 1d. If anything, it is the *KMT1A-MLLT3* alone accelerating the leukemia the most. A statistical test comparing the primary to each mutation's secondary should be done or the authors should state that the leukemias are not accelerated.

Response: Yes, it's correct that *KMT2A-MLLT3* alone accelerate disease the most as emphasized in Fig. 1d and in the text. As suggested, we performed a statistical test between the primary and secondary leukemias which showed that all secondary leukemias had a significantly shorter disease latency as compared to their primary counterpart. These results have been added to the Supplementary information (Supplementary Figure 4d) and we have modified the text on page 6 line 112 to clarify that there was a significant difference.

4. What do authors want to conclude by showing smaller fraction of cells co-expressing *FLT3N676K* and *KMT2A-MLLT3* in SPL compared to in BM in primary recipients? Fig 2c. why did the authors not show %GFP+ in the same organ in primary and secondary transplantation?

Response: In Figure 2c we want to show how *KMT2A-MLLT3+FLT3^{N676K}* co-expressing cells expand in secondary recipients. Since splenocytes from primary leukemias and not BM were used for the secondary transplantations, we show the % of *KMT2A-MLLT3+FLT3^{N676K}* cells in splenocytes and compare it to their frequency in BM in the secondary recipients at the

time of sacrifice. We have now clarified that primary leukemia splenocytes were used as source for the secondary transplantations on page 7, line 154 which refers to Figure 2c. We have also included the comparison between primary BM and secondary BM as a new supplementary figure (Supplementary Figure 5k), showing the same tendency.

5. The section about MIF has several issues. To which data/figure authors refer when they stated mice expressing subclonal FLT3N676K succumb earlier to the disease than other mice lacking this mutation? At the beginning of this section, authors tried to explain the advantages of subclonal expressing FLT3N676K compared to those lacking this mutation, but later on they went on to show that Mif is higher in all leukemias, including the ones co-expressing other activating mutations (FLT3-ITD and NRASG12D)., Fig 5c could have been strengthened with a western blot.

Response: When stating that mice expressing subclonal *FLT3*^{N676K} succumb earlier to disease, we are referring to Figure 2b which shows that not only mice with an activating mutation in all leukemia cells have accelerated disease onset, but that also mice with subclonal (<50% of *KMT2A-MLLT3+FLT3*^{N676K} cells) succumb earlier to disease, as compared to mice harboring *KMT2A-MLLT3* alone. We have now added a reference to this figure in the MIF section (page 14, line 326). We have also modified the text describing our hypothesis to make the reasoning clearer (page 6 line 121, and page 14, lines 326-332).

As correctly pointed out, MIF is expressed in all leukemias with activating mutations (FLT3 ITD or N676K and NRAS G12D), but also in leukemias lacking an activating mutation (but at lower levels). Our hypothesis is that activating mutations, whether present at high or low MAF, likely contribute to leukemogenesis in several ways, one of which could be that they influence levels of secreted factors (Supplementary Figure 10a) that in turn affect survival and/or growth of the leukemia cells. Given the high expression of MIF in all mouse leukemias with activating mutations (Figure 5a,b) and in human infant *KMT2A-AFF1* positive ALL (Figure 5c), together with the known role of MIF as a suppressor of TP53 and its role in CLL and in AML (Hudson, J. D. et al., *J. Exp. Med.*, 1999; Fingerle-Rowson, G. et al., *Proc. Natl. Acad. Sci. U.S.A.*, 2003; Abdul-Aziz, A. M. et al., *Cancer Res*, 2017), we reasoned that MIF was a likely candidate to exert such pro-leukemic effects. However, Mif is likely is not the only factor that has the ability to influence leukemogenesis, but herein we show that Mif is able to influence leukemia cell survival (Fig. 5d and Supplementary Fig. 11b). As described above, we have modified the text in the Mif section, page 14, lines 326-332 to clarify this.

Further, to complement the proteomic analysis, we have performed a western blot on primary leukemia cells with dominant clones of activating mutations, confirming higher levels of Mif in the presence of an activating mutation (FLT3-ITD, FLT3-N676K, and NRAS-G12D). Given that Mif is a secreted factor, the bands are weak but the tendency clear. These results have been added as Supplementary Figure 11a.

6. Supplemental Fig 1c should be moved to the BaF3 experiment or after the gene expression profiling.

Response: This figure verify that the correct cDNA is expressed and that they activate expected downstream signaling pathways and we therefore believe that this figure is best suited in the beginning of the result section and would like to keep it in its current position.

7. The authors should include a normal control for the level of p-p38 in supplemental Fig 3 to say it is high in all leukemias.

Response: This sentence has been re-written to “All leukemias displayed similar phosphorylation levels of p38²⁵ (Supplementary Fig. 3).” on page 6 line 108.

8. Page 6, Supplemental Fig 5a, which experiment corresponds to which ratio of KMT2A-MLLT3+ FLT3N676K: KMT2A-MLLT3 alone. It is confusing how the data are presented. It does not look like it is the ratio of cells in Supplemental Fig 5a. There should be a schematic explanation.

Response: To clarify which experiment that corresponds to which ratio, we have added the ratio and how the ratio was calculated next to the flow cytometry plot for each experiment in Supplementary Figure 5a. Note that the figures have been rearranged to better establish the link between each experiment and their respective survival curves (Supplementary Figure 5c is now 5b and Supplementary Figure 5b is now S5c).

9. Authors should provide more detail for how the fish plots were determined in the methods.

Response: We have added a section describing how the fish plots were done in the materials and methods section, page 25, lines 580-584.

10. In Fig S5h, the axis should be %GFP+mCherry+

Response: Supplementary Figure 5h shows the fraction of GFP⁺ cells among mCherry⁺ leukemia cells. However, we noticed that we had described this incorrectly in the figure legend and have corrected this.

Reviewer #2: Major Comments/Questions

In leukemias associated with MLL translocations, concomitant mutations in KRAS, NRAS and FLT3, among others, occur frequently. The current study follows on from Dr Andersson’s prior work (Andersson et al., 2015, Nat. Genetics) which characterized the mutational landscape of infant ALL. Here, the authors show (as is well established – see refs 13-20 in the manuscript) that coexpression of signal transduction activating mutations in FLT3/RAS with MLL-AF9 accelerate onset of AML in a mouse model.

In cell mixing transplant experiments (with MLL-AF9 cells), the authors show that MLL-AF9/FLT3 N676K cells (which cycle more rapidly than MLL-AF9 only control cells) contribute to reduced disease latency when either a major clone or a sub-clone. Mice secondarily transplanted with leukemic splenocytes from mice with higher proportions of MLL-AF9/FLT3 N676K cells exhibited shorter disease latencies; the size of MLL-AF9/FLT3 N676K clones mostly expand in secondary recipients versus MLL-AF9 only control clones.

Targeted sequencing of (?) 122 murine leukemias was performed & point mutations in Braf, Cbl, Kras and Ptpn11 were identified in a minority of cases – mutant allele frequency analysis demonstrated some clones to be major & others to be minor sub-clones.

Coexpression of these acquired mutations (although CBL A308T not done) with MLL-AF9 shortens leukemia latency.

RNA sequencing was performed on many of these leukemias (and normal progenitor cells as comparators) and, perhaps unsurprisingly given they cycle faster, leukaemias with co-

expressed activating mutations differentially express specific gene sets such as MYC targets, ribosomal genes and translation genes.

This is clearly a huge amount of work; the manuscript is generally well written and the figures well presented.

It is of interest that mutations frequently found in human MLL leukaemias are spontaneously acquired during the course of leukaemia development in mouse models of the same. It is noteworthy that, as in humans, their role seems to be to accelerate growth of the leukemic clone rather than being absolutely required for transformation or development of full-fledged leukemia.

There are some significant concerns. First, it is hard to discern what is truly novel here: the association of activating mutations in signal transduction pathways with MLL leukemias is well established, likewise the fact that the introduction of these mutations accelerates (through enhancing proliferation) the onset of leukemia. The RNAseq, proteomic and GSEA analyses reveal the results one would expect if two MLL-leukemia clones are compared, one of which is cycling faster than the other. Overall the conclusions presented here seem incremental.

1. An acquired CBL A308T mutation in MLL-AF9 AML cells is argued to provide the clone with the ability to “out compete” an MLL-AF9/FLT3 N676K sub-clone, although the equivalent mutation has not been reported in human malignancy. Functional studies in BaF/3 cells are shown, although these are difficult to interpret.

Response: The specific mutation likely interferes with the binding of target substrates, thereby leading to prolonged activation of tyrosine kinases (Sanada M. et al., *Nature*, 2009; Sargin B. et al., *Blood*, 2007). It has previously been described that in order for Cbl mutants to cause IL-3 independent growth in BaF/3 cells, the mutants need to be co-expressed with a wild-type receptor-tyrosine kinase (i.e. Fernandes M. S. et al., *J. Biol. Chem*, 2010; Reindl C. et al., *Clin. Cancer Res.*, 2009; Polzer H., et al., *Exp. Hematol.*, 2013). We have now clarified this in the section regarding the BaF/3 Cbl experiment on page 9 lines 199-204.

2. Mass spectrometry/proteomics was also performed, although the results of these analyses do not seem to be discussed in detail (e.g. where is the list of identified proteins?) and may perhaps more usefully be omitted.

Response: We would like to keep the proteomic analyses since they conform the gene expression analyses; we have added the lists of identified proteins and their abundance (Supplementary Table 8).

3. There follows extensive gene set enrichment analysis, some of which is difficult to understand.

Response: We have modified the text on pages 13-14 describing the gene set enrichment results and we hope that this makes the analyses performed clearer.

4. Finally, the authors note that MLL-AF9/FLT3 N676K AML cells express higher levels of macrophage migration inhibitory factor (Mif) transcripts by comparison with control MLL-AF9 AML cells. They also state that protein levels are higher (line 312) but the accompanying Figure 5B shows no significant difference.

Response: Its correct that while Mif transcript levels reached statistical significance, the MIF protein levels of did not, but showed a trend towards higher MIF expression in leukemias with activating mutations as compared to those lacking such a mutation. To verify the mass-spectrometry data, we performed western blots, showing that *KMT2A-MLLT3* leukemias with an activating mutation (n=6, two each of FLT3^{ITD}, FLT3^{N676K}, and NRAS^{G12D}) generally expressed higher levels of MIF as compared to leukemias lacking an activating mutation (n=2). Taken together, MIF was more highly expressed at both the mRNA and protein level in *KMT2A-MLLT3* leukemias with an activating mutation, although a variation was observed among samples. This data has been added as Supplementary Fig. 11a and we have also added a section in the Materials and Methods, pages 23-24.

5. An experiment is shown which implies that Mif might be able to sustain survival of growth factor deprived AML cells *in vitro*, although the panel (Figure 5E) & legend do not make it easy to understand precisely what experiment has been performed. No *in vivo* data are provided.

Response: We have updated the legend to Figure 5d page 44 lines 1091-1092, to clarify that this experiment was performed without IL3, a cytokine that gives strong proliferative signals to the *KMT2A-MLLT3* leukemia cells.

In addition, we performed two different *in vivo* experimental approaches to address the role of Mif in leukemogenesis. In the first approach, we tried to retrovirally overexpressed Mif together with *KMT2A-MLLT3* to determine its effect on survival and in the second approach, we determined if MIF had the ability to affect the number of leukemia initiating cells (LICs) using a well-established model of enriched dsRed+ *KMT2A-MLLT3* LICs that was established by Benjamin Ebert (Miller, P. G. et al., *Cancer Cell*, 2013). In this model GMPs transduced with *KMT2A-MLLT3* were serially transplanted in order to enrich for LICs.

Overexpression of Mif in leukemic cells did not show a significant difference in survival of recipient mice. However, this is possibly due to insufficient levels of Mif *in vivo* as western blots of primary leukemia cells revealed no clear difference between Mif “overexpressing”- and control (*KMT2A-MLLT3* only) leukemia cells. Thus, even though control experiments in BaF3 cells showed that the construct expresses Mif, it does not produce high levels of MIF *in vivo*. Given that we were unable to confirm overexpression of Mif in these transplantations we feel that we are unable to draw decisive conclusions from this experiment and have decided not to include this data in the revised version of the manuscript.

In the second approach, LIC enriched cells were treated *ex vivo* with or without MIF for 3 days and cells were then transplanted into sublethally irradiated recipients. MIF treatment *ex vivo* showed that MIF preserved the number of LICs and this translated to a significantly shorter survival for mice that received MIF treated cells as compared to no cytokine control. In the latter model, GMPs transduced with *KMT2A-MLLT3* were serially transplanted to enrich for LICs. Such cells were treated *ex vivo* with or without MIF for 3 days and cells were then transplanted into sublethally irradiated recipients. These data have been added to the manuscript in the result section page 15, lines 343-351 and in Figure 5e,f and Supplementary Figure 11b-f.

6. The argument that a minor MLL-AF9/FLT3-N676K AML clone shortens AML latency by secreting factors that alter the behaviour of the major clone is not well supported by the data here; it seems more probable that it shortens latency by contributing to the total number of leukaemia cells in the mouse.

Response: We have no indications that activating mutations affect the total number of leukemia cells in the mice since there was no significant difference between the groups in the total number of harvested leukemia cells, with the exception of mice receiving NRAS-G12D, that had fewer number of BM cells (New Supplementary Fig. 1i). Although we cannot rule out that there are other causes than secreted factors, such as direct cell-cell contact, we believe that secretion of stimulating factors (with Mif being one of them), is a likely explanation.

7. An additional point is the extent to which the experimental system models human disease. Seemingly in the previous work (Andersson et al., 2015, Nat. Genetics) there were no FLT3 mutations associated with MLL-AF9 (Figure 2b). Instead the two FLT3 N676K mutations were found as major clones in MLL-AF4 cases (Table S20). The biology of MLL-AF4 versus MLL-AF9 is quite distinct, as evidenced by the great difficulty there has been over the years modelling MLL-AF4 infant ALL (in contrast to MLL-AF9 which readily transforms BM stem and progenitor cells). Might FLT3 N676K have a role specifically in infant MLL-AF4 disease?

Response: In our previous work, Supplementary Table 25 shows “Lesions and mutant allele frequencies of genes within the kinase/PI3K/RAS pathway” across infant and non-infant *KMT2A-R* leukemias, and for infant non-*KMT2A-R* ALLs (Andersson A. K. et al., *Nat. Genet.*, 2015). We identified a total of 8 *FLT3* mutations across the *KMT2A-R* patients, 7 of these were considered activating. Four of the seven mutations affected N676K and was thus the most commonly mutated amino acid in *FLT3*. As correctly pointed out, among infant cases both N676K mutations were found in *KMT2A-AFF1* rearranged cases and both were in the dominant leukemia clone. However, the two additional N676K mutations were found in the non-infant cohort, both in AML cases harboring a *KMT2A-ELL* or a *KMT2A-MLLT3* fusion, respectively. In the *KMT2A-ELL* case, the N676K mutation was subclonal (MAF 0.25) and for *KMT2A-MLLT3* the *FLT3*-N676K was found as a dominant clone (MAF 0.35). Thus, although one cannot rule out that this mutation has a specific role in infants with *KMT2A-AFF1*, we do not think that the N676K mutation is unique to this subtype, but rather that *KMT2A-AFF1* constituted the largest genetic subtype of cases in the infant group enabling its detection.

8. Lines 119-121 & accompanying Figure S5A; the precise meaning of the Figure in relation to the text is unclear. How does the figure illustrate the experiment described in the text? Please make this clearer.

Response: Supplementary Figure 5a shows the transduction efficiencies for the three performed experiments. We agree that its not intuitive which experiment that corresponds to which ratio and we therefore added the respective ratio and how it was calculated in connection to the flow cytometer charts of the transduction efficiencies across the three experiments.

9. Lines 123-125; not clear to which of the three experiments mentioned on line 119 the authors refer to here. Likewise, for Figure S5c – which graph refers to 1:28, 1:41 and 1:156 cell mixing experiment?

Response: We have modified the text on page 6 line 129 to underscore that the data we refer to are a combined analysis of three experiments (note that we also show data separated per experiment in Supplementary Fig. 5b). In addition, we have noted the ratio (1:28, 1:41 and 1:156) of *MLLT3*-mCherry+*FLT3*^{N676K}-GFP:*KMT2A-MLLT3*-mCherry at each experiment in Supplementary Fig. 5a. Note that the figures have been rearranged to better establish the link

between the respective survival curves and each experiment (Supplementary Fig. 5c is now 5b and Supplementary Fig. 5b is now 5c).

10. Lines 167-170; not clear which murine leukemias exhibited which mutations. Please make this clear. Please state whether these mutations confer constitutive signaling activity.

Response: We have modified the text on Page 8, lines 176-179 to make this clear and we now also clearly state that all but one of the mutations occurred at amino acid positions known to be activating. In addition, to clarify which murine leukemias exhibited which mutations, we have updated Supplementary Table 4 to include the genotype (column B) in addition to the subject ID and mutation.

11. Figure S7A; this is not clear – does this graph show growth of BAF/3 cells without IL3? If not, why do the control cells seemingly die? Also, why include FLT3 transduction in this experiment? If the CBL A308T mutation is truly able to replicate constitutively active RAS pathway signalling, surely it should read out on its own in this assay?

Response: Its correct that Supplementary Figure 7a show Ba/F3 cell growth without IL3 and we have now clarified this in the Figure. Mutations in *CBL* have been described to cause factor independent growth of Ba/F3 cells only when overexpressed with certain receptor tyrosine kinases (mainly class III receptor tyrosine kinases) such as *FLT3*, *KIT*, *PDGFRA*, and *PDGFRB* (Fernandes M. S. et al., *J. Biol. Chem*, 2010; Reindl C. et al., *Clin. Cancer Res.*, 2009; Polzer H., et al., *Exp. Hematol.*, 2013). Cbl mutations likely interfere with the E3 ubiquitin ligase activity of Cbl, binding of its target substrates or inhibit that of wild-type Cbl, thereby leading to prolonged activation of tyrosine kinases (Sanada M. et al., *Nature*, 2009; Sargin B. et al. *Blood*, 2007) which might explain the need of overexpression of a wild-type receptor tyrosine kinase in the BaF/3 system. We therefore used the same approach and overexpressed *CBL*^{A308T} and *FLT3*^{WT} to assess the ability of the *CBL*^{A308T} mutation to target FLT3 for ubiquitination and degradation. The control cells express WT *CBL* and *FLT3* and here, no effect should be seen as wild-type CBL is able to regulate FLT3 signaling through ubiquitin-mediated degradation. We have modified the text on page 9, lines 199-204 to make the rationale clearer.

12. Figure 3g; this is not clear – were there three separate cohorts of mice, each with 3 mice? Please make this clear.

Response: These are only three mice, each with an activating de novo mutation. We have modified the sentence referring to this figure to make this clear on page 10, lines 211-212.

13. Please show the gene names that permit the cluster analysis shown in Figure 4A.

Response: A list of these genes has been added as Supplementary Table 10.

14. Lines 285-6; it is not clear what is meant by “known output genes” and “negative feedback regulators...”. Please clarify. This makes the conclusions presented in lines 285-293 difficult to follow or understand.

Response: These genes have been taken from Pratilas C. A. et al., *Proc. Natl. Acad. Sci. U.S.A.*, 2009. The genes were identified by inhibiting the activity of MEK in BRAF V600E mutant cells and the list is comprised of 52 genes that include downstream target genes,

transcription factors that regulate transformation, and known feedback regulators of ERK signaling including Sprouty gene families. Thus, these genes measure both signaling through the pathway, but also if a tumor has a high expression of feedback regulators, which would indicate that cells are insensitive to normal feedback regulation. The text has been modified on pages 14, lines 310-313 to better explain what these genes represent and why the analysis was performed.

15. Line 294/295 – what is the comparator population here? Please make this clear. It is currently difficult to understand the conclusion here.

Response: Here, we performed gene ontology (DAVID) on genes highly expressed for *KMT2A-MLLT3*+Empty-GFP when compared to *KMT2A-MLLT3+FLT3^{ITD}*, *KMT2A-MLLT3+FLT3^{N676K}*, and *KMT2A-MLLT3+NRAS^{G12D}* (i.e. the downregulated genes in the new Supplementary Figure 8f). The text has been modified on page 14, lines 316-318 to clarify this.

16. Gene names in Figure S10A are too small to read. Please enlarge.

Response: We have enlarged the gene symbols in Supplementary Figure 10a.

17. The discussion largely repeats the findings from the results section and could be substantially shortened.

Response: We have now reduced the discussion substantially.

REVIEWERS' COMMENTS:

Reviewer #1 (Remarks to the Author):

We thank the authors for their detailed response to our reviews. Our concerns have been addressed.

Reviewer #2 (Remarks to the Author):

The significant issues and concerns raised regarding novelty and the incremental nature of the study remain and, inexplicably, have not been addressed at all by the authors in the rebuttal letter.

Specifically, and directly extracting from the previous review:

"It is hard to discern what is truly novel here: the association of activating mutations in signal transduction pathways with MLL leukemias is well established, likewise the fact that the introduction of these mutations accelerates (through enhancing proliferation) the onset of leukemia"

and

"The RNAseq, proteomic and GSEA analyses reveal the results one would expect if two MLL-leukemia clones are compared, one of which is cycling faster than the other. Overall the conclusions presented here seem incremental."

The authors have not made any attempt to address these very significant points.

The argument that MIF secretion from a dominant clone might encourage the growth of other clones remains poorly supported by the data presented. The observation that MIF over expression failed to enhance MLL-AF9 leukemogenesis (an experiment requested by Reviewer 1) is not in keeping with the significant role for MIF claimed by the authors.

Reviewer #3 (Remarks to the Author):

All the concerns have been addressed successfully by revising the manuscript and experiments. Congratulations!

Response to Reviewers Comments and Questions

NCOMMS-17-11808A, “*De novo* activating mutations drive clonal evolution and enhance clonal fitness in *KMT2A*-rearranged leukemia.”

Referrals to pages and lines in the manuscript are based on the submitted Word file with track changes “All Markup”.

Reviewer #1 (Remarks to the Author):

1. We thank the authors for their detailed response to our reviews. Our concerns have been addressed.

Response: We are happy to hear that the concerns have been appropriately addressed.

Reviewer #2 (Remarks to the Author):

1. The significant issues and concerns raised regarding novelty and the incremental nature of the study remain and, inexplicably, have not been addressed at all by the authors in the rebuttal letter.

Specifically, and directly extracting from the previous review:

“It is hard to discern what is truly novel here: the association of activating mutations in signal transduction pathways with MLL leukemias is well established, likewise the fact that the introduction of these mutations accelerates (through enhancing proliferation) the onset of leukemia”

“The RNAseq, proteomic and GSEA analyses reveal the results one would expect if two MLL-leukemia clones are compared, one of which is cycling faster than the other. Overall the conclusions presented here seem incremental.”

The authors have not made any attempt to address these very significant points.

Response: It is correct that several studies have addressed the cooperativity of activating mutations and a *KMT2A*-fusion gene in mice. However, none have previously functionally characterized the *FLT3*^{N676K} mutation and showed that a subclonal activating mutation can enhance leukemia onset. Given the high frequency of subclonal activating mutations in cancer, our data therefore adds to our current understanding of subclonal signaling mutations in leukemia pathogenesis. Also, the spontaneous acquisition of *de novo* activating mutations in this model and the clonal evolution of these mutations in mice goes hand-in-hand with human data and the importance of activated signaling in leukemogenesis. To functionally demonstrate that the *de novo* mutations had a functional role in leukemogenesis, we co-expressed the *Ptpn11*^{S506W} mutation with *KMT2A-MLLT3* showing that *Ptpn11*^{S506W} accelerate AML onset. The *Ptpn11*^{S506W} corresponds to the known hotspot at S502 in human disease, which is known to increase its phosphatase activity, but herein we provide functional proof that the mouse variant cooperate with *KMT2A-MLLT3*.

As for the RNAseq and GSEA analysis, although the gene expression changes induced by a *KMT2A*-fusion gene have been extensively studied, only a single previous study has addressed the gene expression changes that are caused by co-expression of an activating mutation (*NRAS*^{G12V}) and *KMT2A-MLLT3* (Sachs et al., Blood, 2014). They proposed that *NRAS*^{G12V} may post-transcriptionally activate MYB, allowing it to facilitate the self-renewal effects of *KMT2A-MLLT3*, and that *NRAS*^{G12V} was required for leukemia self-renewal, independent of its effects on growth and survival and thereby critical for leukemia maintenance. Herein, we validate and extend these previous results to show that also *NRAS*^{G12D}, *FLT3*^{N676K} and *FLT3*^{ITD} likely has similar effects and enforced the MYB

driven leukemia-maintenance signature (Zuber et al., Genes and Dev 2011). Moreover, the same signatures were enriched in *KMT2A-MLLT3* mice that gain dominant de novo mutations. In addition, we extend these findings by showing that the MYC and MYB signatures were also enriched in infant ALL with activating mutations. Finally, we show that for *KMT2A-MLLT3* leukemia cells that contain *NRAS*^{G12D} or *FLT3*^{N676K}, not only the transcriptional output of the MEK/ERK pathway is enhanced, but known negative regulators of the pathway are also enriched, in line with those cells being resistant towards these inhibitory signals (Pratilas et al., PNAS 2009 and Burgess et al., Ca Cell 2017). To clarify this we have made minor modifications to the GEP result section in the revised version (pages 13-14 lines 365-366, 369, 373, 456, 461).

In conclusion, and in line with the comments by reviewers 1 and 3, we strongly believe that by modelling several defined cooperative mutations *in vivo*, our results both extend previous studies and adds significant new knowledge to the mechanisms underlying *KMT2A*-rearranged leukemia.

2. The argument that MIF secretion from a dominant clone might encourage the growth of other clones remains poorly supported by the data presented. The observation that MIF over expression failed to enhance MLL-AF9 leukemogenesis (an experiment requested by Reviewer 1) is not in keeping with the significant role for MIF claimed by the authors.

Response: As referred to in our previous rebuttal letter, two different *in vivo* experimental approaches were performed to address the role of Mif in leukemogenesis. In the first approach, we retrovirally overexpressed *KMT2A-MLLT3* alone or with Mif, followed by transplantation to mice to determine if Mif had the ability to affect survival. In the second approach, we determined if MIF had the ability to affect the number of leukemia initiating cells (LICs) using a well-established model of enriched dsRed+ *KMT2A-MLLT3* LICs that was established by Benjamin Ebert (Miller, P. G. et al., *Cancer Cell*, 2013). In support of our studies demonstrating an effect of MIF on the survival, but not growth, of primary *KMT2A-MLLT3* mouse leukemia cells *ex vivo* (Figure 5d), the latter model demonstrated that MIF stimulation of dsRed+ *KMT2A-MLLT3* LICs followed by transplantation into sublethally irradiated recipients preserved *KMT2A-MLLT3* LICs as compared to no cytokine control mice, which translated to a significantly shorter survival (Figure 5e-f and Supplementary Figure 11b-f). We agree that overexpression of MIF would be a good experiment, but as explained in the previous rebuttal letter, the retroviral construct did not produce high levels of MIF *in vivo* as demonstrated by Western blot and we therefore feel that we cannot draw decisive conclusions from that experiment.

In conclusion, the experiments performed herein, suggests that MIF has the ability to positively affect survival of both primary mouse *KMT2A-MLLT3* leukemia cells as well as of *KMT2A-MLLT3* LICs, similar to what has been reported in human AML and in CLL (Abdul-Aziz et al, Cancer Res, 2017, Binsky et al., PNAS 2007, Reinart et al, Blood 2013). As clearly stated in the manuscript, we do not believe that MIF is the only cytokine that can exert such pro-leukemic effects (page 17, lines 559-560). In the latest revised manuscript, we made minor modifications on page 15, lines 512 in the result section and on page 17, line 558 in the discussion of Mif.

Reviewer #3 (Remarks to the Author):

All the concerns have been addressed successfully by revising the manuscript and experiments. Congratulations!

Response: Again, we are very happy to hear that the reviewer finds our manuscript satisfactorily revised.